# Decoding a cancer-relevant splicing decision in the *RON* proto-oncogene using high-throughput mutagenesis

Simon Braun[1], Mihaela Enculescu[1], Samarth T. Setty[2], Mariela Cortés-López[1], Bernardo P. de Almeida[3,4], F.X. Reymond Sutandy[1], Laura Schulz[1], Anke Busch[1], Markus Seiler[2], Stefanie Ebersberger[1], Nuno L. Barbosa-Morais [3], Stefan Legewie[1], Julian König[1] & Kathi Zarnack [2]

Mutations causing aberrant splicing are frequently implicated in human diseases including cancer. Here, we establish a high-throughput screen of randomly mutated minigenes to decode the *cis*-regulatory landscape that determines alternative splicing of exon 11 in the proto-oncogene *MST1R* (*RON*). Mathematical modelling of splicing kinetics enables us to identify more than 1000 mutations affecting *RON* exon 11 skipping, which corresponds to the pathological isoform RONΔ165. Importantly, the effects correlate with *RON* alternative splicing in cancer patients bearing the same mutations. Moreover, we highlight heterogeneous nuclear ribonucleoprotein H (HNRNPH) as a key regulator of *RON* splicing in healthy tissues and cancer. Using iCLIP and synergy analysis, we pinpoint the functionally most relevant HNRNPH binding sites and demonstrate how cooperative HNRNPH binding facilitates a splicing switch of *RON* exon 11. Our results thereby offer insights into splicing regulation and the impact of mutations on alternative splicing in cancer.

[1] Institute of Molecular Biology (IMB), Ackermannweg 4, 55128 Mainz, Germany. [2] Buchmann Institute for Molecular Life Sciences (BMLS), Goethe University Frankfurt, Max-von-Laue-Str. 15, 60438 Frankfurt, Germany. [3] Instituto de Medicina Molecular João Lobo Antunes, Faculdade de Medicina da Universidade de Lisboa, Av. Prof. Egas Moniz, 1649-028 Lisboa, Portugal. [4] Departamento de Ciências Biomédicas e Medicina, Universidade do Algarve, Campus Gambelas, 8005-139 Faro, Portugal. These authors contributed equally: Simon Braun, Mihaela Enculescu, Samarth T. Setty. Correspondence and requests for materials should be addressed to S.L. (email: s.legewie@imb-mainz.de) or to J.Kön. (email: j.koenig@imb-mainz.de) or to K.Z. (email: kathi.zarnack@bmls.de)

Alternative splicing constitutes a major step in eukaryotic gene expression. More than 90% of human genes undergo alternative splicing[1,2], which allows the production of distinct protein isoforms with different functionalities[3,4] and plays a critical role in development and tissue identity[5]. Strikingly, tumour suppressor genes and proto-oncogenes are particularly susceptible to splicing defects. Moreover, abnormally expressed splicing factors can have oncogenic properties[6], and changes in alternative splicing contribute to key processes in cancer initiation and progression[7–9]. A detailed characterisation of splicing mechanisms is therefore fundamental to our understanding of human biology and disease.

Splicing is an important step in the maturation of nascent transcripts that comprises excision of introns and joining of exons. During alternative splicing, certain exons can be either included or excluded, thus leading to different transcript isoforms. Splicing is catalysed by the spliceosome, a multi-subunit complex that recognises the 5′ and 3′ splice sites and flanking sequence elements in the pre-mRNA. The latter include the polypyrimidine tract (Py-tract) and the branch point upstream of each exon[10]. In addition to these core splice signals, multiple cis-regulatory elements reside in exons and flanking introns which can be primary RNA sequence elements as well as RNA secondary structures. The recognition of cis-regulatory elements by trans-acting RNA-binding proteins (RBPs) guides the spliceosome and ultimately determines the splicing decision at each alternative exon. Altogether, the information in the pre-mRNA sequence and how it is interpreted by RBPs is commonly referred to as the splicing code[11–13].

Despite many efforts to understand the molecular rules of splicing, our knowledge about cis-regulatory elements and trans-acting factors in most cases remains far from complete. Recent bioinformatic studies aimed to decipher the splicing code by predicting the impact of sequence variants on alternative splicing decisions[14,15]. Moreover, mutagenesis screens were employed to map sequence determinants of alternative splicing. However, these studies were limited to targeted mutagenesis of synthetic reporter constructs or short exonic regions[16–18].

Recepteur d'origine nantais (RON) is a receptor tyrosine kinase encoded by the proto-oncogene MST1R (also referred to as RON). Under normal conditions, the protein is cleaved to form a functional receptor. Skipping of RON alternative exon 11 results in the isoform RONΔ165, which remains as a single-chain protein. Spontaneous oligomerisation of RONΔ165 results in constitutive phosphorylation[19] that promotes epithelial-to-mesenchymal transition and contributes to tumour invasiveness[20–23]. Consistently, RONΔ165 is frequently upregulated in solid tumours, including ovarian, pancreatic, breast and colon cancers[21,24,25]. On the molecular level, previous studies identified a handful of mutations that influence RON exon 11 splicing[26,27]. Moreover, several RBPs were reported to regulate RON splicing[7,26,27]. For instance, heterogeneous nuclear ribonucleoprotein H (HNRNPH; collectively referring to HNRNPH1 and its close paralogue HNRNPH2 which are 96% identical at the amino acid level[28]) was found to repress RON exon 11 inclusion via binding within the alternative exon. While these studies suggested that RON splicing is heavily regulated, most cis-regulatory elements remain unknown.

Here, we establish a high-throughput mutagenesis approach to comprehensively characterise the regulatory landscape of RON exon 11 splicing. Starting from a library of almost 5800 randomly mutated minigenes, we employ a mathematical model of the splicing kinetics to detect more than 1000 point mutations that significantly affect RON alternative splicing. Importantly, the deduced single mutation effects correlate with the splicing levels in cancer patients bearing the same mutations. Moreover, we

comprehensively characterise how HNRNPH acts as a key regulator of healthy and pathophysiological RON splicing by recognising multiple cis-regulatory elements in a cooperative fashion. Our mutagenesis screening approach promises insights into the splicing effects of mutations in humans and the mechanisms of alternative splicing regulation in general.

## Results

**Random mutagenesis introduces 18,000 mutations**. To systematically study the cis-regulatory sequence elements that control RON alternative splicing, we designed an in vivo screening approach based on random mutagenesis of a splicing reporter minigene (Fig. 1a). The minigene harbours RON exon 11 together with the complete flanking introns and the constitutive exons 10 and 12 (Supplementary Fig. 1a). We confirmed that the minigene gives rise to the same transcript isoforms as the endogenous gene in human HEK293T cells (Supplementary Fig. 1b). Moreover, mutations in a known cis-regulatory element led to increased RON exon 11 skipping as reported previously[7] (Supplementary Fig. 1c). We next amplified the minigene with error-prone PCR to spread mutations randomly across all exons and introns. A 15-nt barcode sequence was introduced downstream of constitutive exon 12 via a randomised sequence in the reverse primer to uniquely identify each mutated minigene variant. Upon vector ligation and amplification, we pooled ~6000 clones into a minigene library (Supplementary Fig. 1d). As an internal reference, the library was supplemented with wild-type (wt) minigene variants that carry distinct barcode sequences but no mutations.

To map the introduced mutations, we sequenced the minigene library with 300-nt paired-end reads and five overlapping amplicons. The 15-nt barcode included in each read pair enabled us to assign and reconstruct the complete sequence of all minigene variants in the library (Supplementary Fig. 2a). Using a custom-tailored analysis pipeline (Supplementary Fig. 2b), we capture a total of 5791 unique minigene variants (see Methods), including 5200 with randomly introduced mutations as well as 591 with the wt sequence (Supplementary Data 1). Mutation calling identified 18,948 point mutations with an average frequency of 3.6 mutations per minigene variant. The mutations are randomly spread across the RON minigene, such that 97% of the positions are mutated at least ten times within the library (average 28 times per position; Supplementary Fig. 2c–e). We validated the accuracy of mutation calling with Sanger sequencing of 59 randomly selected minigene variants, confirming all 169 mutations without additional false positives.

**Targeted RNA-seq quantifies alternative splicing outcome**. To measure the splicing outcome, we transfected the library as a pool into human HEK293T cells where the minigenes are transcribed and spliced. We devised a targeted RNA-seq strategy based on 300-nt paired-end reads, which allows us to assemble the complete sequence of all splice products including the 15-nt barcode sequence that is present in all read pairs (Supplementary Fig. 2f, g). A total of 5598 (97%) minigene variants were captured in all three independent biological RNA-seq replicates (Supplementary Fig. 2h and Supplementary Data 1). From the RNA-seq data, we could reconstruct and quantify 163 distinct splice isoforms. The most abundant isoforms reflect the canonical splicing events, i.e., alternative exon (AE) inclusion and skipping, as well as partial and full intron retention (IR) (Fig. 1b, g and Supplementary Fig. 2g). In addition, we detected non-canonical splicing events at 82 and 71 cryptic 3′ and 5′ splice sites, respectively, which are collectively referred to as 'other' (Fig. 1c). For instance, mutations disrupting the 3′ splice site of the downstream constitutive exon 12 trigger activation of a cryptic AG (marked by one asterisk in

Fig. 1b, g). While the overall abundance of the cryptic isoforms in the RNA-seq libraries is low, they can dominate the splice products of individual minigene variants (Fig. 1b).

For the wt minigenes, i.e., in the absence of mutations, the frequency of the AE inclusion isoform (i.e., the ratio of AE inclusion over the sum of all measured isoforms) shows little variance, supporting the notion that confounding effects of the barcode sequences are negligible (Fig. 1d). In contrast, almost half of the mutated minigenes (2248, 45%) show more than 10% deviation in AE inclusion, suggesting that many introduced mutations strongly affect the splicing outcome (Fig. 1d). As

expected, any mutation within the splice sites of RON exon 11 completely abolishes AE inclusion (Fig. 1d, f). We validated the accuracy of the RNA-seq quantification using individual RT-PCR measurements of the 59 Sanger-sequenced minigene variants (Fig. 1e and Supplementary Data 8). We conclude that the random mutagenesis approach enables precise high-throughput quantification of alternative splicing.

**Linear regression modelling infers single mutation effects.** Since each mutated minigene variant carries several mutations, the measured splicing changes are an overlay of multiple effects.

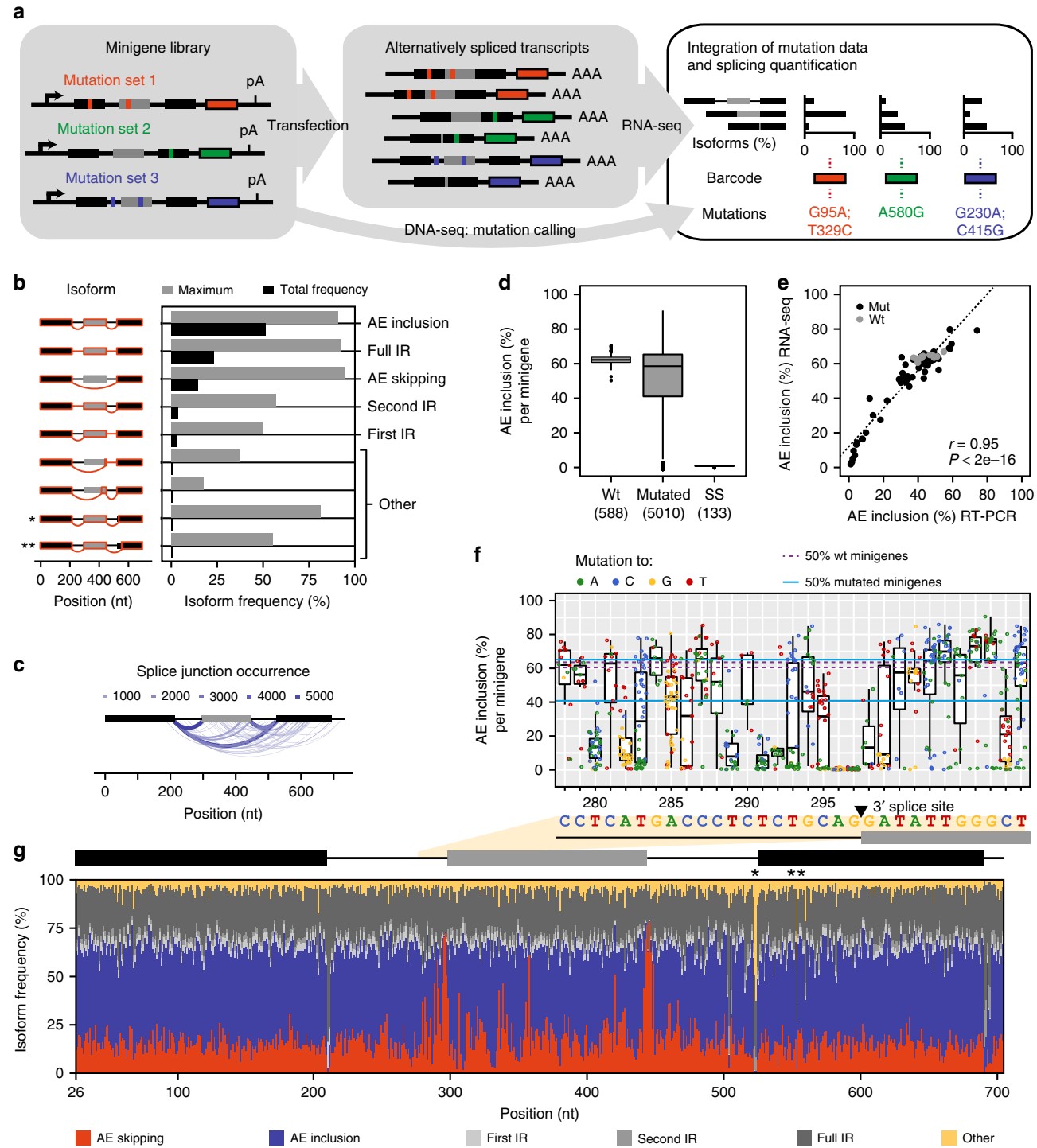

Consequently, a set of minigenes that share a given mutation displays a certain degree of variation in their splicing behaviour (Fig. 1f and Supplementary Data 2). To extract the impact of individual mutations, we made the simplifying assumption that mutations affect splicing independently and derived a linear regression-based mathematical modelling approach. In the linear regression model, the splicing change of each minigene relative to wt is described as the sum of single mutation effects (Fig. 2a). By fitting this model to the measured combined mutation effects, the underlying single mutation effects can be inferred.

To assess whether additivity of mutation effects can indeed be assumed, we analysed a reaction network representing splicing of the RON minigene using kinetic modelling (Supplementary Note 1 and Supplementary Fig. 3a). Model analysis shows that only when we consider splice isoform ratios (i.e., ratios of two measured isoform frequencies), mutation effects do not depend on the presence of other mutations in a minigene. Thus, for splice isoform ratios, mutation effects add up in log-space and a linear regression can be performed. In contrast, at the level of individual splice isoform frequencies (or related metrics such as percent spliced-in, PSI), mutation-induced fold changes depend on the mutational background and are thus not additive in log-space. We directly confirmed the additive behaviour of isoform ratios for mutations that are present as single mutation minigenes and simultaneously occur as combinations in double/triple mutation minigenes (Supplementary Fig. 4).

To integrate the full mutation information available in the data set, we formulated five separate regression models, each expressing the splicing outcome as a ratio of one splice isoform relative to the reference AE inclusion isoform. By simultaneously fitting the complete set of linear equations, each reflecting one minigene, to the experimental data, we were able to estimate 1800 single mutation effects. Based on the regression results, we could infer the frequency of five canonical splice isoforms for each of these single mutations, or combinations thereof (Supplementary Table 1 and Supplementary Data 3). The models fit the data with high accuracy, as judged by the excellent correlation between model fit and experimental data (Pearson correlation coefficient, $r = 0.99$, $P$ value $< 2e-16$; Fig. 2b and Supplementary Fig. 5a, b). This supports our assumption that mutations affect splicing independently and can be described as a sum of single mutation effects (Supplementary Figs. 3b and 4a).

To test for ability of the model to infer novel combined mutations, we employed tenfold cross-validation, in which the model was fitted to 90% of the minigenes and used to predict the splicing outcome for the remaining 10%. The excellent cross-validation accuracy (Pearson correlation coefficients $r = 0.96-0.97$, $P$ value $< 2e-16$; Supplementary Fig. 6) outperformed alternative regression model variants that were fitted directly to the measured splice isoform frequencies (Supplementary Note 2 and Supplementary Fig. 7a, b). The inference power of the model for novel single mutation effects was assessed by separately leaving out one of >500 single-mutation minigene variants and fitting the model to the remaining data, or to subsets, in which further occurrences of the considered mutation were left out. This procedure revealed that the inference error for a single mutation effect decays with increasing occurrence of a mutation in our data set as $(E \sim 1/\sqrt{occurrence})$ (see Supplementary Note 2, Fig. 2c and Supplementary Fig. 5c). For mutations with occurrences >5 (i.e., present in more than five minigene variants), the estimated standard deviation of the inference error levels around 6%, suggesting that at sufficiently high occurrence the model inference accuracy is close to the experimental variation for wt minigene variants (3% standard deviation). We compared the cross-validation results to a simpler proxy in which single mutations effects are estimated from the median splice isoform frequencies over all minigenes containing a particular mutation. Even though the latter approach should average out the effect of accompanying mutations when enough minigenes are present, the regression model outperforms the median-based estimation across all occurrence levels (Fig. 2c and Supplementary Fig. 5c, d).

To independently validate the modelling results, we generated 26 minigene variants with individual mutations for which the model predictions substantially differed from the simpler median-based estimation of single mutations effects. Using RT-PCR to assess splicing outcomes, we find a strong correlation with the splice isoform frequencies inferred by the model (Fig. 2d, Supplementary Fig. 4b and Supplementary Data 8). The gain in accuracy by the model is particular apparent for mutations with a low frequency, i.e., appearing in only few minigenes. We conclude that the regression model offers a reliable method to quantify the impact of single mutations on RON alternative splicing.

**Numerous positions contribute to RON alternative splicing.** Using the model inference for HEK293T cells, we find a total of 778 mutations that significantly alter the frequency of at least one isoform (henceforth called splicing-effective mutations; >5% change in isoform frequency, 5% false discovery rate, FDR; Fig. 2e–g, Supplementary Fig. 8 and Supplementary Table 2). At the 5′ splice site of RON exon 11, we observe a good correlation between AE inclusion levels and in-silico-predicted splice-site

**Fig. 1** High-throughput mutagenesis screen provides quantitative splicing information across the RON minigene. **a** High-throughput detection of splicing-effective mutations. Mutagenic PCR creates mutated minigene library (left) that gives rise to alternatively spliced transcripts (middle). Mutations and corresponding splicing products are characterised by DNA and RNA sequencing, respectively, and linked by unique 15-nt barcode sequence in each minigene (coloured boxes). Black and grey boxes depict constitutive and alternative exons, respectively. **b** Nine most frequent isoforms found in HEK293T cells. Bar diagram shows total frequency in RNA-seq library (black) and maximal frequency for any individual minigene variant (grey). Asterisks mark non-canonical isoforms from cryptic 3′ splice site usage upon mutations at positions marked in **g**. AE, alternative exon, IR, intron retention, other, non-canonical isoforms. **c** Occurrence of distinct splice junctions in HEK293T cells. Line thickness and colour represent number of minigene variants producing a given junction (only junctions accounting for ≥1% of all junctions for a given minigene). **d** Boxplot showing distribution of AE inclusion frequencies (as % of all isoforms) for all wild-type (wt) and mutated minigenes and a subset with mutations in splice sites (ss) of RON exon 11. Boxes represent quartiles, centre lines denote 50th percentile, and whiskers extend to most extreme values within 1.5× interquartile range (IQR). **e** Validation of AE inclusion frequencies for 59 randomly selected minigene variants. Scatterplot compares the RNA-seq quantification to semiquantitative RT-PCR for individual minigene variants in HEK293T cells. r, Pearson correlation coefficient and associated P value. **f** Mutational landscape around the 3′ splice site of RON exon 11. Boxplot of AE inclusion frequencies in HEK293T cells for all minigenes with mutation at indicated positions (x-axis). Box representation as in **d**. Colours illustrate inserted nucleobase (see legend). Blue and purple lines indicate IQR of AE inclusion frequencies for all mutated and wt minigenes, respectively. Sequence of wt RON minigene given below. **g** Isoform frequencies arising from mutations along RON minigene. Stacked bar chart shows median frequency of six isoform categories for all minigenes with mutation at a given position. Average of three biological replicates in HEK293T cells. Asterisks highlight positions where mutations lead to non-canonical isoforms depicted in **b**

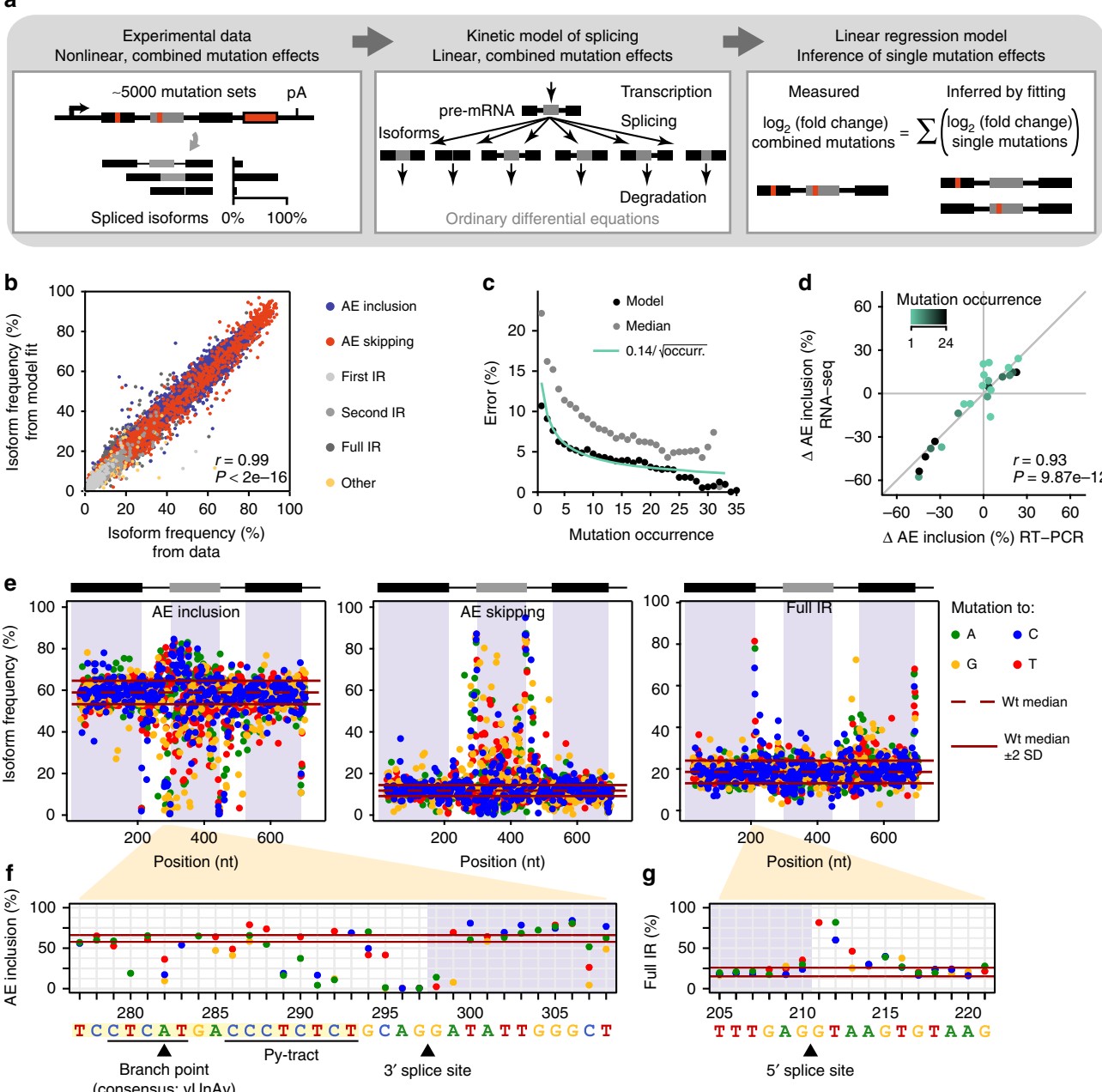

**Fig. 2** A linear regression model determines more than 1900 single mutation effects. **a** Model-based inference of single mutation effects. Isoform quantifications from RNA-seq for 5598 unique minigene variants, each harbouring multiple mutations, are used as input. A kinetic model of splicing reactions reveals that splice isoform ratios show linear mutation effects, irrespective of other mutations. A linear regression model is used to infer single mutation effects in a system of 5010 linear equations, one per mutated minigene variant. **b** Regression model describes experimental measurements with high correlation (Pearson correlation coefficient $r = 0.99$, $P$ value <2e−16). Scatterplot shows frequencies of distinct splice isoforms (see legend; separately shown in Supplementary Fig. 5a) for combined mutations calculated from fitted model against one biological replicate (see Supplementary Note 2). **c** The model more accurately infers frequently occurring single mutation effects. Cross-validation by separately excluding single-mutation minigenes (and permutations of other minigenes containing this mutation). Inference is expressed as standard deviation of inference error in AE inclusion (y-axis) and analysed for different permutations containing mutation at different frequencies (x-axis). Inference power of model (black dots) matches theoretical expectation (green line) and outperforms median-based estimation (grey dots; see Supplementary Note 2). **d** Experimental validation of model-inferred single mutation effects. Semiquantitative RT-PCR measurements of AE inclusion (other isoforms in Supplementary Fig. 4b) for targeted single-mutation minigenes that were not used for model fitting. Discrepancies between model and data appear if mutation infrequently occurs in the library (colour-coded). $r$, Pearson correlation coefficient and associated $P$ value. **e** Model-inferred landscapes of 1747 single mutation effects on AE inclusion, AE skipping and full IR in HEK293T cells. Each mutation effect is indicated as a coloured dot (inserted nucleobase, see legend). Red lines indicate median (dashed) ± 2 standard deviations (SD; solid) for wt minigenes. **f**, **g** Zoom-in landscapes of single mutation effects on AE inclusion around 3′ splice site of *RON* exon 11 (**f**) and on full IR around 5′ splice site of constitutive exon 10 (**g**). Black lines and arrowheads mark splicing signals, including branch point, polypyrimindine tract (Py-tract) and splice sites. Visualisation as in **e**

strength upon mutation[29] (Spearman correlation coefficient $r = 0.89$, $P$ value $= 2.36e-08$; Supplementary Fig. 9a). In contrast, predictions for 3′ splice site strength capture the effects of Py-tract composition, but fail to detect branch point mutations and other sequence contributions (Spearman correlation coefficient $r = 0.62$, $P$ value $= 4.02e-07$; Supplementary Fig. 9b). As expected, transitions between pyrimidines within the Py-tract upstream of *RON* exon 11 act neutrally, whereas transversions into purines reduce inclusion, illustrating that the screen allows to discriminate base-specific effects (Fig. 2f). Consistent with the exon definition model of splicing, we find that disrupting the 5′ splice site of constitutive exon 12 (not spliced in the minigene context) also changes AE inclusion (Fig. 2e, Supplementary Table 2 and Supplementary Data 4), underlining that flanking constitutive exons can distally influence alternative splicing[11,30].

Notably, 91% of all positions within *RON* exon 11 (134/147 nt) harbour at least one splicing-effective mutation, revealing that the alternative exon is densely packed with *cis*-regulatory elements (Fig. 2e and Supplementary Fig. 8). Moreover, neighbouring positions or even different base substitutions in the same positions often affect different isoforms or change splicing in opposite directions (e.g., regions 404–429 nt or 565–567 nt, respectively, in Supplementary Data 4). The resulting patterns likely resemble footprints of the RNA sequence specificity of the interacting RBPs (see below) or RNA secondary structures. In addition to disrupting existing *cis*-regulatory elements, some mutations may also generate new elements, which further increases the complexity of the observed regulatory landscape.

The widespread occurrence of splicing-regulatory effects in *RON* exon 11 highlights that the majority of exonic positions mediate splicing regulation and thus harbour a second layer of information beyond their protein-coding function. As previously described[16], splicing-regulatory effects occur with similar frequency and effect sizes for synonymous and non-synonymous mutations (Supplementary Fig. 9e, f). Moreover, we detect substantial effects in the flanking introns and constitutive exons (50–82% splicing-effective positions per region; Supplementary Table 2). Albeit less frequent, the splicing-effective mutations within introns show comparable effect sizes to those in the alternative exon (Supplementary Fig. 9d). Globally, mutations in and around the alternative exon primarily affect the AE inclusion and skipping isoforms, whereas mutations in the downstream constitutive exon strike a balance between AE inclusion and full intron retention (Supplementary Fig. 8).

In line with a pathological relevance, we find that splicing-effective positions within introns are more conserved than non-effective positions, evidencing an evolutionary selection pressure towards maintaining the splicing-effective positions[31] (Supplementary Fig. 9c). In contrast, within exons both splicing-effective and non-effective positions show high conservation but no difference, likely reflecting constraints on amino acid composition that may overrule conservation of splicing signals. A total of 135 (25%) of splicing-effective mutations within the three exons are synonymous with respect to the encoded RON protein and would hence not be interpreted as potentially deleterious variants when considering protein sequence only. Importantly, our results clearly indicate that albeit preserving the protein sequence, such synonymous mutations may contribute to disease by changing alternative splicing patterns[32,33].

**Splicing-effective positions are mutated in human cancers.** Since altered *RON* splicing is involved in cancer progression[21,25], we repeated the splicing measurements in the human breast cancer cell line MCF7. Compared to HEK293T cells, the wt minigene shows lower AE inclusion in MCF7 cells, supporting a

shift towards the pathophysiological state (Supplementary Fig. 1b). Nevertheless, the measured mutation effects are highly consistent between both cell lines, underlining the robustness of our screening approach (Pearson correlation coefficient $r = 0.96$, $P$ value $= 2.2e-16$; Fig. 3a).

In order to address the physiological relevance of the mutations, we compared our data to the Catalogue of Somatic Mutations in Cancer (COSMIC). Out of 33 COSMIC entries within the region of the *RON* minigene, 20 coincide with splicing-effective mutations and seven of these are synonymous with respect to the encoded RON protein (Fig. 3b). It is thus conceivable that their splicing-regulatory function rather than their protein-coding role is involved in cancer progression. Prompted by this observation, we analysed patient data from The Cancer Genome Atlas (TCGA) (https://cancergenome.nih.gov/) to investigate *RON* splicing in human cancers. We identified 153 patients, from 19 different cohorts (representing different cancer types), that carry mutations in the *RON* minigene region specifically in their tumour samples, but not in their matched normal samples (Supplementary Data 5). We next quantified the difference in *RON* exon 11 splicing (in PSI), per cohort, between tumour samples of mutation-bearing and non-bearing patients. Strikingly, we observe a good correlation between *RON* splicing changes in mutated TCGA tumour samples and the single mutation effects determined by our approach (Pearson correlation coefficient $r = 0.62$, $P$ value $= 4.8e-05$; Fig. 3c). Strongest *RON* exon 11 skipping associates with a splice site mutation (G297A; identified in a patient with thyroid carcinoma, THCA). Of note, the second largest effect is found for mutation G370T (head–neck squamous cell carcinoma, HNSC), which introduces a missense mutation at the level of the encoded protein (Fig. 3d, see Discussion). The correlation between our screen and the TCGA data is reduced if these two strongest sites are removed from the analysis (Pearson correlation coefficient $r = 0.27$, $P$ value $= 0.12$; Fig. 3c), most likely because the remaining effects are weaker and compromised by experimental variation. In conclusion, our high-throughput screen recapitulates strong in vivo splicing changes in human cancer patients.

**cis-Regulatory elements in *RON* are targeted by multiple RBPs.** In MCF7 cells, a total of 1022 mutations across the minigene affect *RON* alternative splicing, pointing towards the presence of multiple *cis*-regulatory elements (Supplementary Data 6). We used the ATtRACT database[34] to identify putative RBP binding sites, thereby predicting RBPs that recognise these *cis*-regulatory elements. In order to focus on sites that are actively involved in splicing regulation, we retained only RBP motifs if at least 60% of the positions therein showed a mutation effect on at least one splice isoform (referred to as splice-regulatory binding sites, SRBS). The analysis recovers two previously reported *cis*-regulatory elements in the alternative and the downstream constitutive exon that are targeted by HNRNPH[26] and SRSF1[21], respectively. In total, we identify 76 potential RBP regulators (Fig. 3e and Supplementary Fig. 10), suggesting that *RON* splicing is extensively controlled by multiple RBPs. To prioritise among them, we overlaid our data with a large-scale knockdown (KD) screen which tested the KD effect of 31 RBPs from our list on *RON* exon 11 splicing in HeLa cells[35]. Notably, 17 of these RBPs showed a substantial impact on *RON* splicing, with HNRNPH and SRSF2 being the strongest repressor and activator, respectively (Fig. 3e).

In a complementary approach, we investigated the expression of 190 RBPs which were identified as putative regulators of *RON* splicing by our ATtRACT analyses and/or by the published RBP KD screen[35] using matched RNA-seq data sets for 4514 TCGA

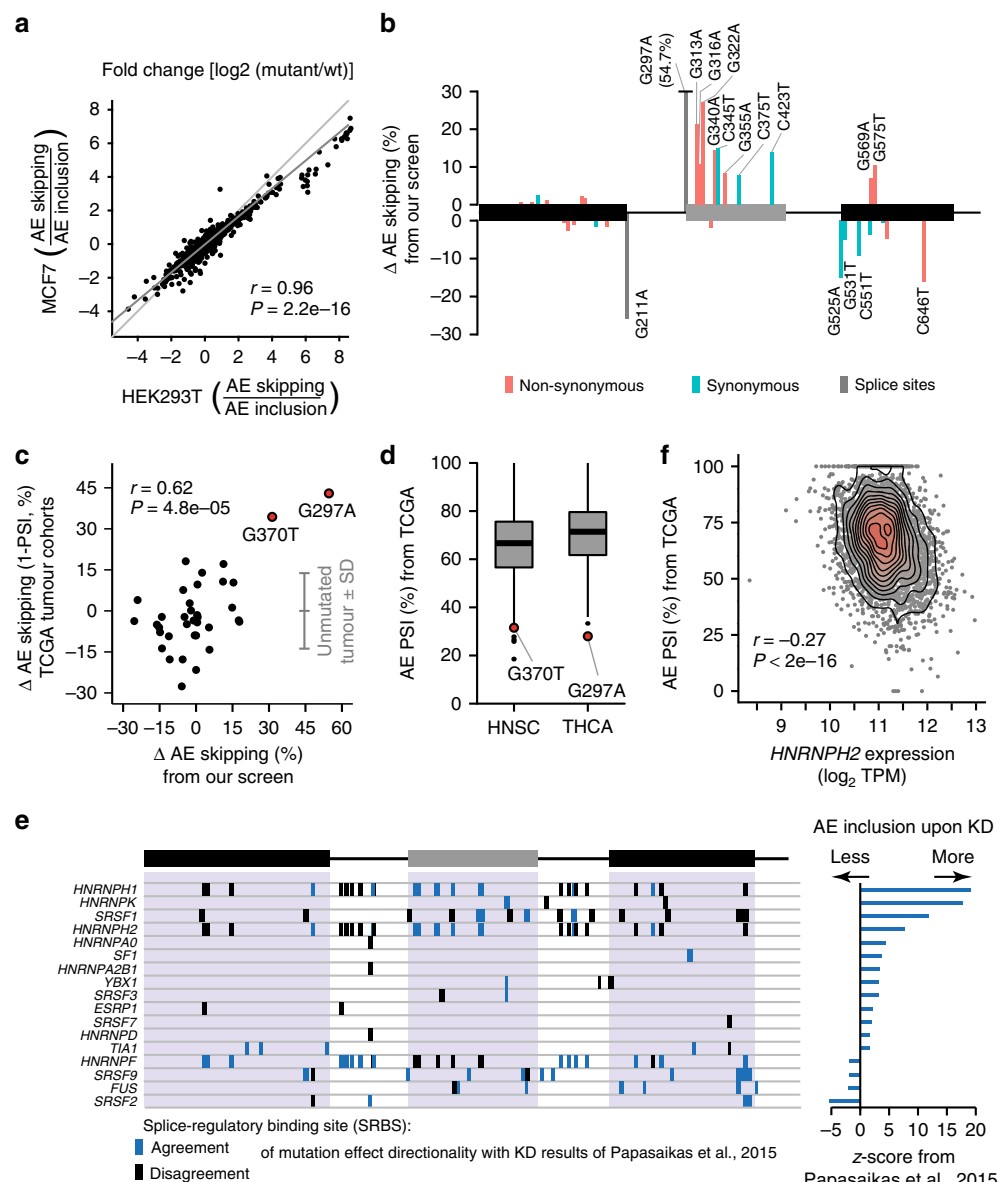

**Fig. 3** Mutation effects are recapitulated in human cancer patients and point to regulatory RNA-binding proteins. **a** Model-inferred single mutation effects are consistent between HEK293T and MCF7 cells. Scatterplot compares changes in splice isoform ratios for mutations along *RON* minigene. Light and dark grey lines correspond to diagonal and linear regression, respectively. *r*, Pearson correlation coefficient and associated *P* value. **b** Somatic mutations in cancer can cause significant splicing effects. Bar diagram displays changes in AE inclusion for 33 mutations from COSMIC database. Mutations with significant effect on AE skipping are labelled (*n* = 16). Orange, blue and grey indicate non-synonymous, synonymous and splice site mutations, respectively. Mutation G297A extends beyond visualised range. **c** Mutation effects from our screen recapitulate altered splicing in cancer patients. Scatterplot compares *RON* AE skipping between mutated and non-mutated TCGA tumour samples (percent spliced-in, PSI) from 117 cancer patients (with 36 different mutations in 14 cohorts) to mutation-induced change in AE skipping in MCF7 cells. Grey lines indicate mean and standard deviation of unmutated tumour samples. *r*, Pearson correlation coefficient and associated *P* value. *r* = 0.27, *P* value = 0.12 without two mutations with strongest impact (G370T and G297A). **d** Tumour samples from mutation-bearing patients show strong *RON* exon 11 skipping. Boxplot summarises *RON* exon 11 inclusion (PSI) in head–neck squamous cell carcinoma (HNSC) and thyroid carcinoma (THCA) cohort. Box represents quartiles, centre line denotes 50th percentile and whiskers extend to most extreme data points within 1.5× interquartile range. Tumour samples with mutations G370T and G297A are labelled. **e** In silico predictions for RNA-binding proteins (RBPs) identify splice-regulatory binding sites (SRBS; predicted binding sites that show substantial mutation effects, see Methods). Boxes indicate SRBS for 17 putative RBP regulators that overlap with published data on *RON* exon 11 splicing upon RBP KD[35]. Colour code indicates whether majority of mutation effects within SRBS agree with direction of published RBP KD effect (*z*-scores; right panel). **f** *HNRNPH2* shows strongest correlation of expression levels with *RON* exon 11 splicing across 27 tumour cohorts. Density scatterplot shows *HNRNPH2* expression (in transcripts per million, TPM) and *RON* exon 11 PSI across all TCGA tumour samples. *r*, Spearman correlation coefficient and associated *P* value

cancer patient samples from 27 different cancer types. We detect 140 RBPs whose transcript levels significantly correlated with *RON* exon 11 inclusion (FDR for Spearman correlation <5%; Supplementary Fig. 11a–c and Supplementary Data 7). Compared to all annotated RBPs or all protein-coding genes, the 190 pre-selected

RBPs significantly enriched among the most highly correlated (gene set enrichment analysis, *P* value = 0.04 or 0.003, respectively).

Strikingly, the strongest association in the TCGA data set is observed for *HNRNPH2*, whose expression shows a significant negative correlation with *RON* exon 11 inclusion (Spearman

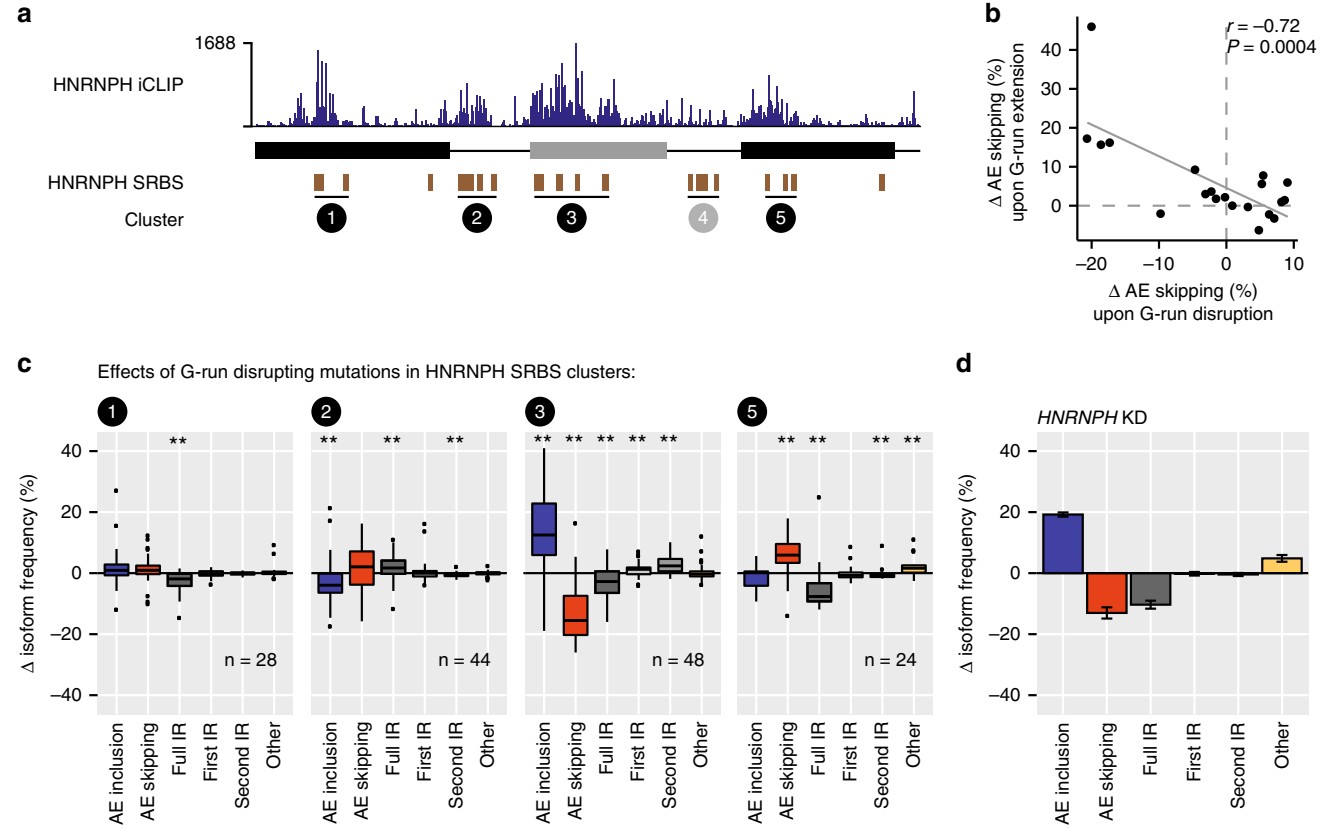

**Fig. 4** HNRNPH controls *RON* exon 11 splicing via multiple intronic and exonic binding sites. **a** HNRNPH iCLIP validates HNRNPH binding to predicted splice-regulatory binding sites (SRBS). Bar diagram shows the number of HNRNPH crosslink events from HEK293T cells on each position along the wt *RON* minigene. HNRNPH SRBS (brown boxes) were assigned to five SRBS clusters (circled numbers). iCLIP data show HNRNPH binding to four out of the five SRBS clusters. **b** Extending or disrupting G-runs results in opposite splicing effects. Scatterplot shows inverse correlation of median changes in AE inclusion (average of three biological replicates) of all mutations per SRBS that extend or disrupt the G-run (see Methods) with linear regression line. *r*, Spearman correlation coefficient and associated *P* value. **c** Exonic and intronic HNRNPH binding exerts distinct effects on *RON* exon 11 splicing. Boxplots summarise the change in frequencies of each isoform in MCF7 cells (mean, *n* = 3) for all G-run-disrupting mutations within HNRNPH SRBS of different clusters (circled numbers). Box represents quartiles, centre line denotes 50th percentile and whiskers extend to most extreme data points within 1.5× interquartile range. Number of considered mutations for each cluster given below. *P value < 0.05, **P value < 0.01, one-sample Wilcoxon test against population mean of zero. **d** Bar diagram shows the changes in isoform frequency of wt *RON* minigenes upon *HNRNPH* KD in MCF7 cells. Error bars indicate standard error of the mean from three biological replicates

correlation coefficient $r = -0.27$, *P* value $< 2e-16$; Fig. 3f). This behaviour is consistent with the previously described function of HNRNPH as a repressor of *RON* exon 11 inclusion[26]. Differentiating into the 27 TCGA cohorts, we observe a significant correlation in 11 individual cancer types (FDR for Spearman correlation < 5%; Supplementary Table 3), all negative, suggesting that HNRNPH2-mediated repression of *RON* exon 11 commonly occurs in human cancers. Notably, we find a similar association in RNA-seq data of 24 healthy human tissues from the Genotype-Tissue Expression (GTEx) Project[36] (Spearman correlation coefficient $r = -0.12$, *P* value $= 5.7e-11$; Supplementary Fig. 11d). Comparing GTEx and TCGA samples, we observe consistently lower *RON* exon 11 inclusion levels in the tumour samples (mean PSI 76% vs. 67%, *P* value $< 2.2e-16$, Mann–Whitney–Wilcoxon test), supporting an increased expression of the constitutively active RONΔ165. Accordingly, *HNRNPH2* expression is increased in cancer (mean transcripts per million [TPM] 57.88 vs. 46.29, *P* value $< 2.2e-16$; Mann–Whitney–Wilcoxon test). Together, these observations suggest that HNRNPH is a major determinant of *RON* alternative splicing in healthy human tissues and cancer.

**HNRNPH binding can both activate and repress *RON* splicing.** Within the *RON* minigene, we predict 22 SRBS for HNRNPH (Fig. 4a), all of which harbour the G-rich sequences (G-runs) recognised by HNRNPH[37]. The HNRNPH SRBS occur across all transcript regions and arrange into five clusters, each containing at least three SRBS (clusters 1–5, Fig. 4a). Individual-nucleotide resolution UV crosslinking and immunoprecipitation (iCLIP; Supplementary Fig. 12a) in HEK293T cells confirms that endogenous HNRNPH significantly binds at the predicted HNRNPH SRBS clusters (Fig. 4a and Supplementary Fig. 12b), with the exception of cluster 4. The strongest iCLIP signal locates in the alternative exon (Supplementary Fig. 12b).

Consistent with HNRNPH's sequence preference towards G-runs, mutations within the binding sites show opposing impact when either disrupting or generating G-runs in the RNA sequence (Fig. 4b). The direction and the most susceptible isoform depend on the position of the HNRNPH SRBS. Most prominently, mutations within SRBS cluster 3 in the alternative exon promote inclusion (Fig. 4c). A similar splicing pattern is observed for the wt *RON* minigene upon *HNRNPH* KD (Fig. 4d),

indicating that cluster 3 plays an important role in HNRNPH-mediated repression of *RON* exon 11. Mutations in the intronic clusters 2 reduce AE inclusion, whereas mutating cluster 5 in the downstream constitutive exon 12 leads to decreased intron retention, accompanied by increased AE skipping. These observations cumulate into a complex regulatory scenario, in which HNRNPH acts via multiple binding sites that have activating or repressing effects on *RON* splicing.

**Synergy analysis identifies predominant HNRNPH sites.** In order to identify which sites are most relevant for HNRNPH-dependent regulation, we tested the splicing response of the minigene library upon *HNRNPH* KD. We hypothesised that mutations that either weaken or reinforce an HNRNPH binding site would display positive or negative synergy with the *HNRNPH* KD. For instance, a reduced KD response compared to the wt minigene would be expected if an important HNRNPH binding site is compromised by a mutation (negative synergy).

In order to test this idea, we performed siRNA-mediated *HNRNPH* KD in MCF7 cells expressing the minigene library and used targeted RNA-seq to measure the splicing outcome. As previously reported[26,35], *HNRNPH* KD results in a strong increase in *RON* exon 11 inclusion for both wt and mutated minigene variants in the library. In line with synergy, a subset of minigene variants reproducibly show a weaker or stronger KD response compared to the remainder of the library. For instance, minigenes harbouring mutations G305A or G310A within cluster 3 consistently show elevated control AE inclusion levels, but a reduced KD response, suggesting that HNRNPH regulation is partially abolished due to these mutations (Fig. 5a).

To comprehensively identify synergistic interactions, we again turned to linear regression modelling and inferred the single mutation effects in control and KD conditions (Fig. 5b). We then calculated a *z*-score, in which the difference in the KD effect between wt and mutant is normalised by the experimental variation of the wt minigenes. Using our model based on isoform ratios (Supplementary Note 3 and Supplementary Fig. 13), we estimate that the *HNRNPH* KD on average has a 2.4-fold effect on the AE skipping-to-inclusion isoform ratio. Importantly, this effect is largely independent of the mutational background and hence the AE inclusion frequency which a minigene exhibits under control conditions (Fig. 5b, right). The exception are very strong mutations that on their own completely abolish splicing, i.e., prevent the KD from having additional measurable effects (Supplementary Fig. 7f, g). In contrast, at the level of individual splice isoform frequencies, the KD effect of all mutations strongly depends on the starting isoform level and thereby introduces systematic biases (Fig. 5b, left, and Supplementary Fig. 7e). Since such biases can be minimised by modelling splice isoform ratios, our approach allows to reliably estimate synergy.

By modelling the splice isoform ratios, we derive landscapes of synergistic interactions between *HNRNPH* KD and distinct mutations in the *RON* minigene sequence (Fig. 5c, Supplementary Fig. 12c and Supplementary Data 6). For the vast majority of point mutations, no significant synergistic interaction is observed (1428 out of 1786, 80%; Supplementary Table 2). Importantly, 354 mutations (20%) in 278 positions show significant synergy for at least one splice isoform (|*z*-score| > 2, adjusted *P* value < 0.001, Stouffer's test). These are significantly enriched in the SRBS in cluster 3 in the alternative exon, in which 64% of mutations (93% of positions) display synergy with *HNRNPH* KD (Fig. 5d and Supplementary Fig. 12c, d). This observation suggests that

the SRBS in the alternative exon are most relevant for HNRNPH-dependent regulation. This is further supported by the fact that 42% of the strongest synergistic interactions that affect AE skipping (|*z*-score| > 5) fall into SRBS cluster 3 (Fig. 5c). Consistent with the known sequence preference of HNRNPH, we observe that the disruption of G-runs in cluster 3 leads to a weaker KD response (negative synergy; Fig. 5d and Supplementary Fig. 14). Instead, synergistic interactions at clusters 1 and 5 in the constitutive exons frequently reinforce HNRNPH-dependent regulation by creating new or extending existing G-runs, leading to a stronger-than-average KD response (positive synergy; Supplementary Fig. 14). Hence, while the HNRNPH SRBS clusters outside *RON* exon 11 do not prevail under the tested conditions, they can become more important when HNRNPH binding for these sites is increased.

In order to validate the functional relevance of SRBS cluster 3, we generated ten minigene variants with individual point mutations disrupting G-runs within HNRNPH SRBS from the five clusters (Supplementary Data 8), and tested their splicing under *HNRNPH* KD conditions using semiquantitative RT-PCR. Indeed, single mutations in cluster 3, for which the model had inferred the strongest synergistic interactions, almost completely cancel out the KD response (Fig. 5e). In contrast, minigenes with mutations in other clusters still respond to the *HNRNPH* KD, in agreement with their less pronounced synergistic interaction with HNRNPH. In summary, the synergy analysis allows to link an RBP to its functionally most relevant *cis*-regulatory elements.

**Cooperative HNRNPH binding establishes a splicing switch.** We find that individual G-run-disrupting point mutations within the alternative exon (e.g., G305A, G331C and G348C) are sufficient to almost completely abolish the response to *HNRNPH* KD (Fig. 5e). This suggests that the corresponding SRBS cooperatively recruit HNRNPH. In line with this notion, our linear regression model which does not consider cooperativity, on average provides a worse fit to minigenes containing simultaneous mutations in two HNRNPH SRBS in the alternative exon (Supplementary Fig. 15e). In order to test for interdependent HNRNPH binding, we repeated the HNRNPH iCLIP experiments in the context of mutated *RON* minigenes harbouring point mutations within three different SRBS of cluster 3. In line with cooperative binding, the resulting drop in HNRNPH crosslinking is not limited to the site of the point mutation, but spreads to several further SRBS in *RON* exon 11 (Fig. 6a).

Cooperative HNRNPH binding to multiple SRBS would imply that HNRNPH regulates splicing with a steep, sigmoidal dose–response curve. To test this, we performed gradual *HNRNPH* KD and *HNRNPH1* overexpression experiments, in which we transfected MCF7 cells with increasing amounts of HNRNPH-specific siRNA and *HNRNPH1* overexpression construct, respectively. Notably, we observe a switch-like splicing response of *RON* exon 11 from the minigene as well as the endogenous *RON* gene. Indicative of strong cooperativity, we find that the dose–response curves can be described by high Hill coefficients for the endogenous *RON* gene (nH = 17.4, confidence interval (CI) [10.8,35.2]) as well as the transfected wt *RON* minigene (nH = 13.8, CI [10.4,17.7]; Fig. 6b and Supplementary Figs. 15a–d, 16). Consistently, we observe that *HNRNPH2* shows the steepest regression slope among the 190 RBPs tested for expression correlation in the TCGA data (Figs. 3f and 6c). Even though *HNRNPH2* expression in the TCGA data is not variable enough to reach plateaus in splicing, the steep slope further supports a switch-like behaviour of *RON* exon 11 inclusion.

Based on these observations, we conclude that *RON* exon 11 splicing is extensively regulated via multiple interdependent HNRNPH binding sites that exert strong cooperativity. This enables switch-like splicing with small changes in HNRNPH concentration causing large changes in splicing (Fig. 6d), potentially explaining why *HNRNPH* expression is a strong predictor of *RON* exon 11 splicing in cancer cells.

## Discussion

Systems approaches combined with mathematical modelling are required to fully comprehend the complex regulation of alternative splicing. Our work builds on previous approaches to measure the effect of mutations in defined regions of splicing-reporter minigenes[16–18,38]. Central to our analytical framework is the mathematical splicing model which allows us to predict the

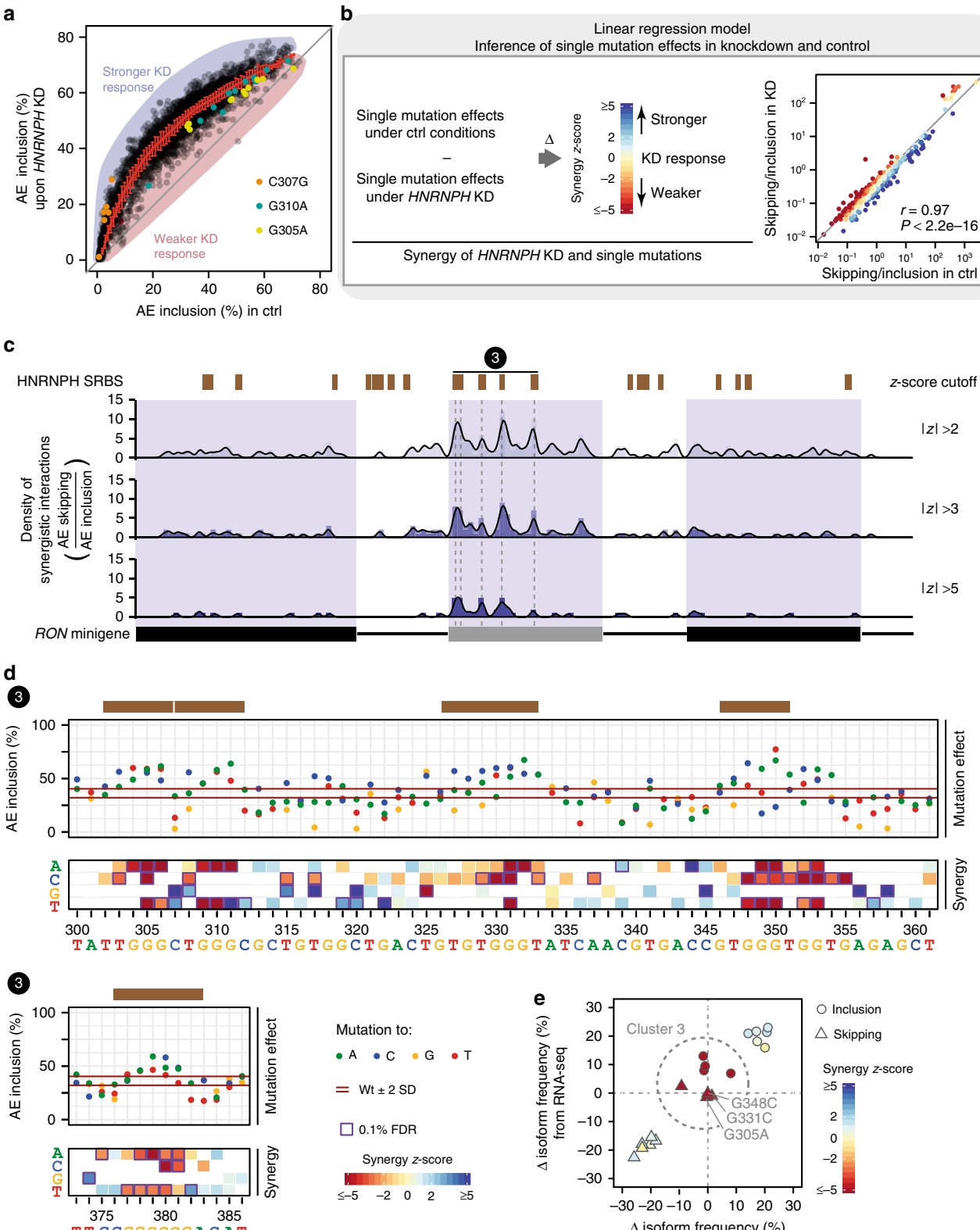

effects of individual mutations based on measurements of combined mutation effects. We employ linear regression modelling to disentangle these effects, and validate the predictive power of this approach using cross-validation, targeted single mutations and by relating *RON* mutations to splicing outcomes in cancer patients.

To formulate the linear regression model, we investigated how mutation effects cumulate in minigenes exhibiting several mutations. We termed a mutation effect linear if a mutation induces the same fold change in splicing irrespective of the mutational background (i.e., other mutations being present). If linearity holds true, the mutation effects add up in log-space and a linear regression can be performed to infer single mutation effects from the measured combined mutations. Using a kinetic model reflecting *RON* alternative splicing, we found that splice isoform ratios show the desired linear behaviour (Supplementary Note 1 and Supplementary Fig. 7e). Accordingly, the isoform ratio-based regression model fitted the complete minigene library with high accuracy. In line with our approach, Rosenberg et al. quantitatively modelled the contribution of randomised *k*-mer sequences in 25-nt regions of a synthetic minigene using an additive model that was based on the AE inclusion-to-skipping ratio[18]. In contrast, direct linear regression using the splice isoform frequencies decreases the accuracy and inference power of the model (Supplementary Note 2 and Supplementary Fig. 7b), possibly indicating nonlinear interactions between mutations at this level. Therefore, care needs to be taken when interpreting the interplay of mutations and/or other perturbations directly based on the abundance of certain splice isoforms (e.g., percent spliced-in/PSI, or equivalent metrics), as each perturbation shifts the operating point of the system. As a global trend, we observe that minigenes showing inclusion frequencies around 50% are most sensitive to perturbations such as *HNRNPH* knockdown (Fig. 5a). However, after transformation to isoform ratios, *HNRNPH* knockdown elicits linear, context-independent changes (Fig. 5b). Thus, isoform ratios are superior when analysing the interplay of multiple treatments or mutations, while isoform frequencies are essential for judging the physiological impact of splicing changes.

Our conclusion that combined mutations can be accurately described as a linear combination of single point mutations implies that synergistic interactions between mutations have only a minor impact on *RON* splicing outcomes. Intriguingly, our data suggest that even simultaneous mutations in two *cis*-regulatory elements mostly elicit linear, independent effects: across more than 100 minigenes containing two or more simultaneous

mutations in any HNRNPH SRBS, 93% of the splice isoform frequencies can be explained within 5% deviation from the measured value using the linear regression approach (Supplementary Fig. 15e). Thus, the goodness of fit of this subset is comparable to the complete minigene population, suggesting that *cis*-regulatory elements in many cases act independently on *RON* alternative splicing.

Despite most mutations acting independently, we observe cooperative effects for adjacent HNRNPH binding sites in the alternative exon (see below). Such nonlinear effects are not captured by our model, but are in line with previous work showing that splicing-relevant mutations can amplify each other in combination, thereby showing cooperative interactions[11,16]. That such nonlinear effects are more prevalent in a previous screen by Julien et al.[16] may result from the fact that their study systematically screened double mutations in close vicinity. However, when relating the goodness of fit of our linear regression model to the nearest distance between two splicing-effective mutations, we found no clear effect of the mutation proximity on the fitting error (Supplementary Fig. 7c, d). This suggests that also nearby mutations typically act independently of each other and agrees well with results from a recent saturation mutagenesis study of a 51-nt region in the alternatively spliced exon of the *WT1* gene[17]. Since our data set does not exhibit enough coverage to comprehensively detect cooperative interactions of nearby mutations, it remains possible that adjacent sites mutually influence each other, whereas distal *cis*-regulatory elements act independently. Such a scenario would be consistent with a local assembly of ribonucleoprotein complexes that act as independent regulatory units.

Our high-throughput mutagenesis screen uncovers a highly complex *cis*-regulatory landscape, with >80% of all positions affecting *RON* alternative splicing. Within this set, we recover mutations in all previously identified *cis*-regulatory elements[7,21,26,27,39]. Within the alternative *RON* exon 11, we find that 91% of all positions show a significant impact on *RON* splicing. Conceptually, these splicing-effective mutations either disrupt existing *cis*-regulatory elements at the RNA sequence or structure level or generate novel elements, thereby further increasing the complexity of *RON* splicing regulation. Even though the newly generated *cis*-regulatory elements do not occur under normal conditions, they may be relevant in cancer when mutations accumulate. A similar density of effective mutations was also reported for *FAS* exon 6[16]. Our study demonstrates that other than previously suggested, such a densely packed

---

**Fig. 5** Synergistic interactions highlight the functionally most relevant HNRNPH binding sites. **a** Minigene variants are differentially spliced upon *HNRNPH* knockdown (KD). Scatterplot compares AE inclusion under control (ctrl) and *HNRNPH* KD conditions for all minigene variants. Average behaviour illustrated by running mean and standard deviation (red). Shadings schematically highlight stronger/weaker-than-average KD response. Minigenes with mutations C307G, G310A and G305A in HNRNPH splice-regulatory binding sites (SRBS) cluster 3 are highlighted. **b** Quantification of synergistic interactions by linear regression modelling. Single mutation effects are determined separately for ctrl and *HNRNPH* KD using linear regression and subtracted to estimate KD responses compared to wt (*z*-score based on standard deviation of wt minigenes, colour-coded; see Supplementary Note 3). Right graph shows model-inferred AE skipping-to-inclusion isoform ratios of single mutations in ctrl vs. KD. Regression line indicates average KD effect. Consideration of isoform ratios, as compared to isoform frequencies (**a**), leads to linearisation of KD response in line with predictions of kinetic splicing model (Supplementary Fig. 3a). **c** Model-inferred synergistic interactions accumulate in *RON* exon 11. Bar diagrams quantify significant synergistic interactions affecting AE skipping-to-inclusion isoform ratio using different *z*-score cutoffs in adjacent 5-nt windows. Line indicates density in 5-nt sliding window. Splice sites ± 2 nt were excluded. Predicted HNRNPH SRBS (brown) are given above. **d** Mutations in HNRNPH SRBS cluster 3 lead to increased AE inclusion and reduced *HNRNPH* KD response. Dot plots (top) display single mutation effects (inserted nucleobase, see legend) on AE inclusion (mean, $n = 3$). Red lines indicate median isoform frequency of wt minigenes ± 2 standard deviations (SD). HNRNPH SRBS (brown) are given above. Heat maps (bottom) show *z*-scores as measure of synergy (mean, $n = 3$) per inserted nucleobase. White or grey fields indicate mutations that were not present or filtered out, respectively (see Methods). Purple boxes highlight significant synergistic interactions (0.1% FDR). **e** Consistent with strong synergistic interactions (colour-coded), mutations in cluster 3 almost completely abolish the *HNRNPH* KD response in MCF7 cells. Scatterplot compares model-inferred estimates with semiquantitative RT-PCR measurements of AE inclusion (circles) and AE skipping (triangles) upon *HNRNPH* KD (mean, $n = 3$) for ten targeted mutations in HNRNPH SRBS (Supplementary Data 8)

arrangement is not exclusive to short exons such as *FAS* exon 6[16] (63 nt) or *SMN1* exon 7[38] (54 nt), as *RON* exon 11 (147 nt) is around the average length of human cassette exons[40].

A major advance of our study is that we detect *cis*-regulatory elements along the entire minigene, including introns and flanking constitutive exons. In the constitutive exons, the effect sizes are generally smaller, possibly due to an accumulation of partially redundant exonic splicing enhancers (ESEs) that ensure constitutive splicing[13]. Notably, we find that mutations not necessarily trigger intron retention, but can also specifically swap between AE inclusion and skipping (such as T581G or G686C; Supplementary Data 6). These distal effects agree with previous

observations from positional splicing maps (RNA-maps), showing that RBP binding at flanking constitutive exons can regulate the inclusion of neighbouring alternative exons[41].

Our ATtRACT analysis together with a previous study[35] suggest that splicing of *RON* exon 11 is controlled by a multilayered network of at least 17 *trans*-acting RBPs. Many of these RBPs are linked to multiple binding sites in different regions, further increasing the regulatory complexity. Importantly, significant correlations in our TCGA analyses suggest that many of the regulatory relationships are functional in humans in vivo. A multilayered regulation of alternative splicing events has also been suggested by a recent high-throughput screen for RBP KD

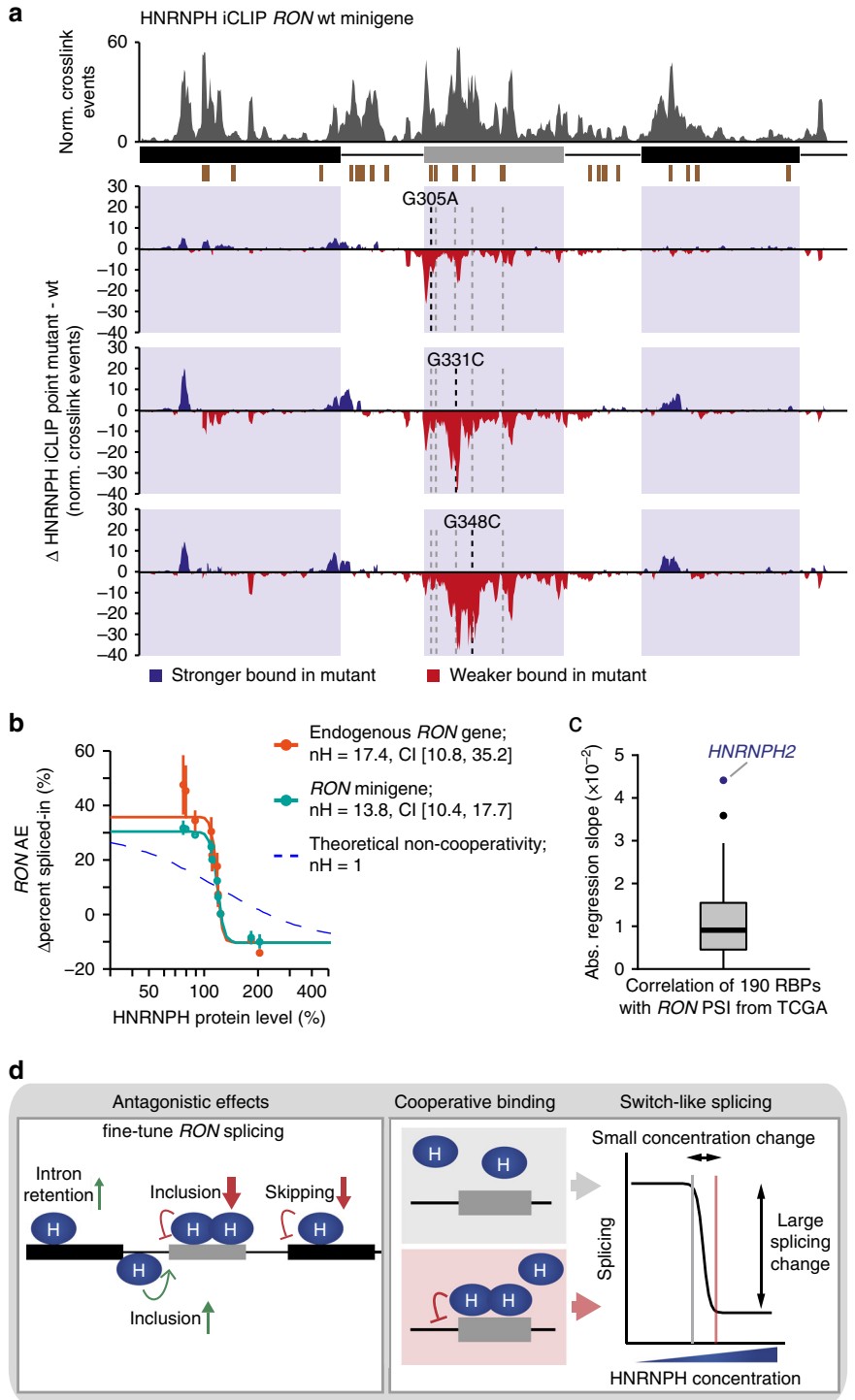

effects[42]. We extend beyond this view by directly identifying synergistic interactions between sequence mutations and RBP KD.

Our study highlights HNRNPH as a key regulator of *RON* exon 11 splicing. Using mutational analysis and iCLIP, we demonstrate that it acts via five clusters of intronic and exonic SRBS that antagonistically affect *RON* splicing (Fig. 6d). In line with previous global splicing maps[37,43,44], we observe that HNRNPH binding in the alternative exon represses AE inclusion, whereas binding in the flanking introns increases AE inclusion. Synergy analysis pinpoints the SRBS in the alternative exon as the functionally most relevant sites. We speculate that this interwoven arrangement of antagonistic SRBS may allow to fine-tune *RON* splicing. More generally, tightly regulated exons might benefit from modulating not just one but several competing splicing reactions in order to achieve an optimal adjustment of alternative splicing under changing physiological conditions.

Other than previously suggested for intronic HNRNPH sites[44], we find strong indications for cooperative binding of HNRNPH to multiple SRBS within the alternative exon. One possible mechanism for the observed cooperativity would be oligomerisation of HNRNPH via its glycine/tyrosine (GY)-rich domain. It was recently shown that other hnRNP proteins form multimeric assemblies via GY-rich domains to regulate splicing[45]. Moreover, HNRNPH binding sites might fold into G-quadruplex structures[46–49], which could contribute to the observed cooperativity through sequestering and simultaneously releasing G-runs. The cooperative HNRNPH binding renders *RON* splicing sensitive to individual mutations in HNRNPH SRBS or to small changes in HNRNPH protein expression.

Extensive changes in alternative splicing are characteristic for many human cancers[50], and it has been estimated that about half of all synonymous driver mutations change splicing[51]. The skipping of *RON* exon 11 results in a ligand-independent, constitutively active variant of the encoded RON receptor tyrosine kinase, RONΔ165[20]. Contrary to initial reports[52], we and others detect RONΔ165 expression also in healthy human tissues[21], suggesting that the encoded protein could play a role under physiological conditions. However, overexpression of RONΔ165 was shown to trigger increased cell motility and invasive tumourigenesis[20]. Consistent with this oncologic potential, abnormal RONΔ165 accumulation has been described in breast and colon cancers, among others[25].

With the help of our mutagenesis screen, we identify many mutations that trigger a strong skipping of *RON* exon 11. Importantly, the mutation effects in our screen are reflected in cancer patients bearing the same mutations. Several of the mutations are synonymous with respect to the encoded protein,

suggesting that mutation-induced splicing changes can have deleterious impact in cancer[51,53–55]. In addition, we also identify numerous non-synonymous mutations that have a strong, and in some cases a surprising, impact on splicing regulation: for instance, the nonsense mutation G370T found in a head–neck squamous cell carcinoma (HNSC) patient also triggers *RON* exon 11 skipping (Fig. 3c, d). Intriguingly, this splicing change inverts the physiological consequence of the mutation, as the majority of mature *RON* transcripts will exclude the mutated exon and thereby translate into a constitutively active rather than a prematurely truncated RON protein.

Due to their prevalence in cancers, altered *RON* isoforms represent a promising target for intervention[56]. For instance, clinical trials assessed the therapeutic potential of monoclonal antibodies targeting RON to block the binding of its ligand MSP (MST1)[57] (ClinicalTrials.gov Identifier: NCT01119456; antibody RON8, Narnatumab, ImClone; phase-I discontinued). However, tumours expressing the constitutively active isoform RONΔ165 can specifically escape this kind of therapies, as this protein no longer requires ligand-dependent activation[24]. A detailed knowledge of mutations that promote this isoform might therefore allow a personalised therapy in the future.

## Methods

**Cloning of the *RON* wt minigene**. To generate the *RON* wt plasmid, a segment of the *MST1R* gene was amplified by polymerase chain reaction using Phusion DNA polymerase (NEB) with the forward primer 5′- CCCAAGCTTTGTGAGAGGCA GCTTCCAGA-3′ and the reverse primer 5′- CAGTCTAGANNNNNNNNNNNN NNNGGATCCGCCATTGGTTGGGGGTAGGGGCTGATTAAAGGTAGG-3′ at 65 °C annealing temperature with human genomic DNA (Promega) as a template (Supplementary Table 4). The 779 bp DNA product was gel-purified with the QIAquick Gel Extraction Kit (QIAGEN) and then digested using *Hin*dIII and *Xba*I restriction endonucleases (NEB). The cut DNA fragment was purified using a PCR purification kit (QIAGEN) prior to ligation into the pcDNA 3.1 (+) vector (Invitrogen). To raise AE inclusion in the *RON* wt minigene comparable to endogenous levels, the first nucleotide of the alternative exon was exchanged to a guanine.

Plasmids harbouring point mutations were generated using the Q5 Site-Directed Mutagenesis Kit (NEB) according to the manufacturer's instructions.

**Mutagenic PCR and library construction**. For the mutagenesis of the *RON* minigene, we used the GeneMorph II Random Mutagenesis Kit (Agilent) according to the manufacturer's instructions. Aiming for an average mutation rate of 3.5 mutations/minigene, three libraries were independently generated and finally fused. To this end, 8 and 4 μg of the unmutated *RON* wt plasmid were amplified with 30 cycles, and 0.8 μg of the unmutated *RON* wt plasmid were amplified with 20 cycles. The primers used to amplify the mutagenic fragments were 5′-CCCAA GCTTTGTGAGAGGCAGCTTCCAGA-3′ (forward primer) and 5′-CAGTCTAG ANNNNNNNNNNNNNNNNGGATCCGCCATTGGTTGGGGGTAGGGGCTGA TTAAAGGTAGG-3′ (reverse primer) (Supplementary Fig. 1a and Supplementary Table 4). The PCR products were purified using the QIAquick Gel Extraction Kit (QIAGEN). Purified DNA was cut with *Hin*dIII and *Xba*I (NEB) restriction endonucleases for 45 min at 37 °C and subsequently purified using a PCR

---

**Fig. 6** Cooperative HNRNPH binding establishes a splicing switch of *RON* exon 11. **a** A single point mutation in an HNRNPH splice-regulatory binding site (SRBS) results in reduced HNRNPH binding in HEK293T cells also at neighbouring SRBS in *RON* exon 11. Bar diagrams show the number of HNRNPH iCLIP crosslink events on the wt *RON* minigene (top) and the difference in normalised crosslink events on wt and mutated *RON* minigenes (mutations G305A, G331C and G348C in different SRBS within cluster 3, marked by dashed lines; bottom) in a sliding 5-nt window along the wt *RON* minigene. HNRNPH SRBS (brown boxes) indicated below. **b** Splicing response to gradual *HNRNPH* KD and overexpression suggests cooperative regulation of *RON* exon 11 by HNRNPH. Scatterplot shows semiquantitative RT-PCR quantifications of *RON* exon 11 inclusion (in percent spliced-in/PSI, Supplementary Fig. 15a, b) from endogenous *RON* gene (orange) and wt *RON* minigene (blue) against corresponding HNRNPH protein levels (Supplementary Fig. 15c, d). Degree of cooperativity is quantified by fitting Hill equation (solid lines) and compared to theoretical fit for non-cooperativity (dashed line). Error bars denote standard deviation of three biological replicates. **c** Steep regression slope for *HNRNPH2* supports cooperative HNRNPH regulation and switch-like splicing of *RON* exon 11. Boxplot shows distribution of regression slopes for expression correlations of 190 RBPs with *RON* exon 11 inclusion in TCGA samples (Supplementary Data 7). Box represents quartiles, centre line denotes 50th percentile and whiskers extend to most extreme data points within 1.5× interquartile range. *HNRNPH2* is highlighted. **d** HNRNPH acts as key regulator of *RON* splicing by recognising multiple *cis*-regulatory elements in a cooperative fashion. Schematic model summarises position-dependent effects of HNRNPH on *RON* exon 11, indicating most strongly effected isoform for each site (left panel). Multiple interdependent HNRNPH binding sites within *RON* exon 11 exert strong cooperative control on the alternative exon, resulting in a splicing switch upon small changes in HNRNPH abundance (right panel)

purification kit (QIAGEN). The digested plasmid DNA and mutagenic fragments were ligated for 5 min at room temperature in a volume of 21 μl containing 50 ng of plasmid and 21 ng of insert (3:1 ratio of insert to plasmid DNA), 10 μl of 2× Quick Ligation Reaction Buffer and 1 μl Quick T4 DNA ligase (NEB). Transformations were carried out via CaCl₂ transformation of *Escherichia coli* DH5-alpha strain with 2 μl of the ligated DNA. Bacteria were plated in low density to allow the formation of similar-sized colonies and determination of the number of transformants by counting of the colonies. Sixteen hours after the transformation, ~2000 colonies per transformation were washed off the plates into lysogeny broth (LB) medium and plasmids were extracted using the Plasmid Plus Midi Kit (QIAGEN). In addition, 200 wt plasmids were generated to be used as a spike-in to the above-mentioned libraries by using the same primers and template wt plasmid but non-mutagenic amplification with Phusion DNA Polymerase (NEB) and the following conditions: 98 °C for 30 s, 30 cycles of [98 °C for 10 s, 61 °C for 20 s, 72 °C for 20 s] and final extension at 72 °C for 5 min. Note that the remainder of wt minigenes in the library represent the proportion of error-free minigenes within the product pool of the mutagenic PCR. Purification, digestion and transformation were performed as described above. Mutagenesis and wt libraries were pooled together to yield a library of ~6000 plasmids. To obtain single plasmids of the library for benchmarking via Sanger Sequencing and validation via RT-PCR, a re-transformation of the library was carried out and plasmids of resulting colonies were extracted using QIAprep Spin Miniprep Kit (QIAGEN).

**Semiquantitative RT-PCR**. Semiquantitative RT-PCR was used to quantify iso-form ratios of individual plasmids and endogenous *RON* mRNA. To this end, reverse transcription was carried out in a volume of 20 μl using 500 ng of total RNA, 1 μl (dT)₁₈ primer (100 μM), 1 μl dNTPs (10 mM) and 1 μl RevertAid reverse transcriptase (Fermentas) by heating 70 °C for 5 min, 25 °C for 5 min, 42 °C for 60 min, 45 °C for 10 min, and 70 °C for 5 min. Subsequently, 1 μl of the cDNA was used as a template for the PCR reaction with the condition as follows: 94 °C for 30 s, 24 cycles (minigene) or 35 cycles (endogenous) of [94 °C for 20 s, 52 °C (minigene) or 62 °C (endogenous) for 30 s, 68 °C for 30 s] and final extension at 68 °C for 5 min. The primers used to amplify the minigene-derived isoforms anneal to the upstream constitutive exon and a region located downstream of the random barcode but upstream of the polyadenylation site: 5′- TGCCAACCTAGTTCCAC TGA-3′ (forward primer) and 5′- GCAACTAGAAGGCACAGTCG-3′ (reverse primer). The primers to amplify endogenously derived isoforms were 5′-CCTGA ATATGTGGTCCGAGACCCCCAG-3′ (forward primer) and 5′-CTAGCTGCT TCCTCCGCCACCAGTA-3′ (reverse primer; Supplementary Table 4). The TapeStation 2200 capillary gel electrophoresis instrument (Agilent) was used for isoform quantification of the PCR products.

**Cell culture and transfection of plasmids and siRNAs**. HEK293T and MCF7 cells were grown in Dulbecco's modified Eagle medium (DMEM) supplemented with 10% foetal bovine serum at 37 °C with 5% CO₂. Standard *HNRNPH* KD was carried out using single small interfering RNA (siRNA) against *HNRNPH*[58] (5′-GGAGCUGGCUUUGAGAGGA[dT][dT]-3′, Sigma-Aldrich) or non-targeting control siRNA (5′-UGGUUUACAUGUCGACUAA[dT][dT]-3′, Sigma-Aldrich) at a final concentration of 20 nM. KD efficiencies were assessed by western blot analyses. For gradual *HNRNPH* KD, the siRNA concentration was varied between 0.05 nM and 10 nM. One day prior to transfection, 2 × 10⁵ HEK293T cells were seeded in a 6-well plate to result with ~20% confluence at the day of transfection. MCF7 cells were seeded 3 days prior to transfection with 0.5 × 10⁵ cells per well of a 6-well plate. The transfection mix was prepared by incubating 3 μl RNAiMax (Invitrogen) with 2 μl siRNA (20 μM) in 200 μl OPTI-MEM (Invitrogen) for 20 min, and the mix was added in a dropwise manner to the cells. For transfection of plasmids 24 h later, a mixture of 2 μg minigene plasmid DNA and 20 μg poly-ethylenimine MW ~2500 transfection reagent (Polysciences, Inc.) in 100 μl OPTI-MEM (Invitrogen) was prepared and incubated for 20 min before it was added to the cells. Cultures were harvested another 24 h later. For the HNRNPH1 over-expression, 4 × 10⁵ MCF7 cells were seeded in a 6-well plate 1 day prior to transfection. Transfection was carried out using Lipofectamine 2000 (Invitrogen) and 1 or 2.5 μg pcDNA 3.1 (+)-HNRNPH1 overexpression construct or pcDNA 3.1 (+) empty vector control. The minigene plasmid was transfected 24 h later as described above and cells were harvested another 24 h later. RNA was extracted using the RNeasy Plus Mini Kit (QIAGEN) according to the manufacturer's protocol. For semiquantitative RT-PCR analysis of splicing isoforms without KD conditions, 7 × 10⁵ HEK293T cells were seeded and transfected the next day with plasmid DNA under the above-mentioned conditions.

No cell line used in this paper is listed in the database of commonly misidentified cell lines maintained by ICLAC. HEK293T (CRL-3216) and MCF7 cells (HTB-22) were purchased from ATCC (Manassas, VA) without further authentication. Cell lines were tested for mycoplasma contamination on a monthly basis.

**Library preparation and high-throughput sequencing**. For preparation of high-throughput RNA sequencing (RNA-seq) libraries, the total RNA obtained from transfected HEK293T cells or MCF7 cells was enriched for mRNA by performing polyA selection of 20 μg of total RNA using Dynabeads® Oligo (dT)₂₅ beads

(Invitrogen) according to the manufacturer's protocol. Reverse transcription was carried out using 500 ng of enriched mRNA under the above-mentioned conditions. To prevent the formation of chimeric amplicons, the libraries were amplified using emulsion PCR[59], with Phusion DNA Polymerase (NEB) and either cDNA derived from polyA-selected RNA in the case of RNA-seq or plasmid DNA of the minigene library in the case of high-throughput DNA sequencing (DNA-seq). To amplify fragments for RNA-seq, the following primers containing Illumina sequencing adaptors were used (Supplementary Fig. 2g): 5′-CAAGCAGAA-GACGGCATACGAGATCGGTCTCGGCATTCCTGCTGAA CCGCTCTTCCGATCTNNNNNNNNNNNGGTTCCACTGAAGCCTGAG-3′ (for-ward primer) and 5′-AATGATACGGCGACCACCGAGATCTACACTCTTTC CCTACACGACGCTCTTCCGATCTNNNNNNNNNATAGAATAGGGCCCT CTAGA-3′ (reverse primer) under the following PCR conditions: 98 °C for 30 s, 15 cycles of [98 °C for 10 s, 56 °C for 20 s, 72 °C for 60 s] and final extension at 72 °C for 5 min. For the DNA-seq library amplification, the same PCR conditions and 18 cycles with different primer combinations were used (Supplementary Fig. 2a and Supplementary Table 4). Following amplification, the DNA-seq PCR products were cleaned using the GeneRead Size Selection Kit (QIAGEN) according to the manufacturer's instructions. Products intended for RNA-seq were purified using Agencourt AMPure XP beads (Beckman Coulter). Purified products were first analysed with the TapeStation 2200 capillary gel electrophoresis instrument (Agilent) and then fluorimetrically quantified using a Qubit fluorimeter (Thermo Scientific). RNA-seq and DNA-seq were carried out on the Illumina MiSeq platform using paired-end reads of 300 nt length and a 10% PhiX spike-in to increase sequence complexity.

**Western blot**. Cell lysates were prepared with modified RIPA buffer (50 mM Tris HCl pH 7.5, 150 mM NaCl, 1 mM EDTA, 1% NP-40, 0.1% sodium deoxycholate, protease inhibitor cocktail; Roche). The following antibodies were used for western blot analyses: rabbit polyclonal anti-HNRNPH, 1:10,000 dilution (AB10374, Abcam) and mouse monoclonal anti-HNRNPA1, 1:10,000 dilution (R4528, Sigma-Aldrich).

**iCLIP experiment and data processing**. We used iCLIP to capture the binding pattern of HNRNPH on the *MST1R* transcript. iCLIP was performed according to a previously published protocol[60]. In brief, the iCLIP libraries were made from HEK293T cells 24 h after transfection of the *RON* wt minigene (in triplicates) or mutated *RON* minigenes carrying point mutations G305A (in triplicates), G331C or G348C (both in duplicates). The cells were irradiated with 150 mJ/cm² UV light at 254 nm. For the immunoprecipitation step, we used 7.5 μg of a polyclonal rabbit anti-HNRNPH antibody from Abcam (AB10374). RNase digestion was performed by adding 10 μl of 1/100 diluted RNase I (Ambion) to the sample of the wt minigene experiment shown in Supplementary Fig. 12a or 1/300 diluted RNase I (Ambion) to each sample of the experiment shown in Fig. 6a (comparison of the iCLIP landscape of the *RON* wt minigene with point mutation minigenes). Reverse transcription was carried out with RT primers listed in Supplementary Table 4. We performed the sequencing on an Illumina HiSeq 2500 for the *RON* wt minigene (51-nt single-end reads) and the *RON* wt/point mutant minigene comparison was sequenced on either Illumina MiSeq or NextSeq 500 with 75-nt single-end reads. Sequencing reads were first filtered for quality in the experimental and random barcode, and then the adaptor sequences were trimmed. Trimmed reads were mapped to the human genome (hg19/GRCh37) using STAR[61] resulting in ~49 million (HiSeq 2500), ~10 million (MiSeq) or ~121 million (NextSeq 500) uniquely mapping reads. In order to quantitatively compare HNRNPH iCLIP data for the *RON* wt and point mutation minigenes (Fig. 6a), crosslink events were normalised to the total number of crosslink events within the minigene region excluding *RON* exon 11. Normalised counts were averaged between replicates, counted into 5-nt sliding windows and then subtracted between conditions to determine differences in HNRNPH crosslinking.

**DNA-seq data processing and mutation calling**. The DNA-seq library was sequenced on Illumina MiSeq (300-nt paired-end) with a total of 40 million reads and analysed with a custom Python pipeline (version 2.7.9: Anaconda 2.2.0, 64-bit; Supplementary Fig. 2b). In detail, we used FastQC (fastqc_v0.11.3; https://www. bioinformatics.babraham.ac.uk/projects/fastqc/) for quality control and Trimmo-matic[62] (version 0.33; parameters HEADCROP:20 SLIDINGWINDOW: 7:10 MINLEN:0) to remove excess sequence and trim low-quality bases (average Phred score < 10 in 7-nt window). After trimming, we filtered for a minimum length of 130 nt (read #1) and 90 nt (read #2). In order to extract the 15-nt barcode (read #1) which assigns the read pairs to an individual minigene variant, we used matchLRPatterns() from the R/Bioconductor package 'Biostrings' to search for the flanking restriction sites (Lpattern = TCTAGA, Rpattern = GGATCC, allowing one mismatch). We only retained read pairs with a Phred score ≥ 30 at all barcode positions. For each minigene variants with at least 640 read pairs, reads were mapped to the sequence of the *RON* wt minigene using NextGenMap[63] (version 0.4.12). A read was reported as mapped if >50% of its bases were mapped, the alignment had an identity >65%, and at least one stretch of 13 bp was identical to the reference. Mutations were called using the HaplotypeCaller tool (version 3.4.0) of the Genome Analysis Toolkit (GATK)[64] with -dt NONE. We recounted

overlapping reads using bam-readcount (https://github.com/genome/bam-readcount) and then manually filtered against single-nucleotide variants (SNV) with low penetrance based on reference (Ref) and alternative (Alt) allele frequencies: (i) Alt/(Alt + Ref) > 0.8, and (ii) (Alt + Ref)/total > 0.5 taking into account all other isoforms. The identified mutations include 18,948 point mutations as well as 608 short insertions and deletions. The latter were taken into account as independent sequence variants in the mathematical splicing model and are provided in addition to the point mutations in Supplementary Data 3. The final library contained 5791 minigene variants, including 591 wt and 5200 mutated minigenes. The accuracy of mutation calling was validated by Sanger sequencing of 59 randomly selected minigene variants, confirming the presence of all 169 GATK-called mutations without further false negatives.

**RNA-seq data processing and splicing isoform quantification.** RNA-seq libraries were sequenced on Illumina MiSeq (300-nt paired-end), yielding 17–22 million reads per sample (Supplementary Table 1), and analysed with a custom Python pipeline similar to DNA-seq (see above; Supplementary Fig. 2f). Briefly, we removed low-quality sequences (average Phred score < 20 in 6-nt window) and extracted the 15-nt barcode (read #1) as described above. Only reads originating from the 5791 minigene variants that were recovered from the DNA-seq library were considered for further analyses. Read pairs for each minigene variant were aligned to the *RON* wt minigene sequence using the splice-aware alignment algorithm STAR[61] (version 2.5.1b), allowing up to ten mismatches without input of prior knowledge of existing splice junctions. Only read pairs conferring splice isoform information (i.e., both mates extended at least 10 nt beyond the constitutive exon boundaries) were kept. Furthermore, all improperly or inconsistently mapped read pairs were removed from the analysis. Read pairs are referred to as improperly mapped if they map with a wrong orientation, while inconsistently mapped read pairs overlap and show a disagreement in their mapping patterns. Finally, only minigene variants which were covered by at least 100 remaining read pairs were used further, resulting in 5697, 5645 and 5623 minigene variants detected in RNA-seq replicates 1, 2 and 3 from HEK293T cells, respectively (Supplementary Fig. 2h and Supplementary Table 1).

**Reconstruction and quantification of splicing isoforms.** For each read pair, the underlying splicing isoform was reconstructed based on the CIGAR strings of the two mates. Isoforms which were supported by <1% of the read pairs or less than two read pairs in any plasmid were removed from the analysis. The frequency of each isoform for each minigene variant was calculated as the number of read pairs supporting this particular isoform in relation to the total read pairs for all detected isoforms for this particular minigene variant. All kept non-canonical isoforms derived from cryptic splice site activation were collected in the isoform category 'other'.

**Dynamic model of splicing.** We modelled the splicing dynamics using a set of ordinary differential equations, in which concentrations of RNA intermediates are determined by production and degradation terms (Supplementary Fig. 3a). The pre-mRNA precursor $x_0$ is produced at a constant rate $c$ and spliced into five splice products with linear kinetics and rates $r_i$. All non-canonical isoforms are included in the model as one additional species produced at rate $r_6$. This leads to $dx_0/dt = c - (r_1 + r_2 + r_3 + r_4 + r_5 + r_6)x_0$. Six additional differential equations describe the dynamics of the canonical (AE inclusion, AE skipping, full IR, first IR and second IR) and non-canonical (other) splice isoforms. The concentration $x_i$ of isoform $i$ is described by $dx_i/dt = r_i x_0 - d_i x_i$, where $d_i$ are RNA degradation rates.

The measured isoform frequencies correspond in the model to the concentration of transcripts $x_i$ normalised by the total RNA concentration. These fractions can be calculated analytically from the steady state of the system (see Supplementary Note 1). As a result, we find that the frequency $p_i$ of a certain isoform $i$ has the form $p_i = K_i/(K_1 + K_2 + K_3 + K_4 + K_5 + K_6)$. Here, the splicing rates $K_j = r_j/d_j$, $j = 1,2,4,5,6$ are the ratios of production and degradation rates for the isoforms involving splicing, and $K_3 = 1 + r_3/d_3$ reflects the sum of the unspliced pre-mRNA ($x_0$) and full intron retention ($x_3$) isoforms, which cannot be discriminated experimentally. Thus, due to normalisation, a change in the production rate of one isoform due to a particular mutation will affect all isoform frequencies, and this effect depends in a nonlinear manner on the values of all splice rates $K_i$ (i.e., on the mutational background). To infer the mutation effects from the data, it is instructive to consider the isoform ratio relative to the inclusion isoform ($p_i/p_1 = K_i/K_1$), as this no longer depends on all splice rates, and relates to $K_i$ in a linear fashion.

**Calculation of single mutation effects by linear regression.** For the estimation of single mutation effects in HEK293T cells, we assumed that the combined log fold changes of multiple mutations on a splice isoform ratio can be expressed as the sum of individual log fold changes (see Supplementary Note 2). One such equation was formulated for each minigene, resulting in a system of 5621–5697 equations for each splice isoform ratio, depending on the amount of minigene variants that were detected in the RNA-seq replicates (Supplementary Table 1).

To support our assumption of additive mutation effects, we analysed how single mutation effects interact in minigenes containing several mutations. To this end, we analysed a subset of mutations that is contained in the library as single mutation

minigenes (~600 minigene variants), and furthermore occur within double/triple mutation minigenes together with other mutations from the list (Supplementary Fig. 4a and Supplementary Table 1). For the majority of these mutations, we observed that the combined mutational effects on the splicing rates $K_i$ are multiplicative, e.g., $K_i(m_1,m_2)/K_i(WT) = K_i(m_1)/K_i(WT)*K_i(m_2)/K_i(WT)$, where $K_i(WT)$, $K_i(m_1)$, $K_i(m_2)$ and $K_i(m_1,m_2)$ are the splicing rates of the wt minigene and of the minigenes including mutation $m_1$ or mutation $m_2$ or both mutations $m_1$ and $m_2$, respectively. In practice, we calculate the mutational effects $K_i(m_1,…m_n)/K_i(WT)$ as a mutation-induced fold change of the splice isoform ratios $p_i/p_1$ (see above). By a log-transformation, the above multiplicative relationship transforms to a linear one that connects the measured cumulative mutation effects with the predominantly unknown single mutation effects (Supplementary Fig. 3a). For the whole pool of measured minigene variants, this constitutes a system of linear equations that can be solved for the single mutation effects in a least-square sense (see Supplementary Note 2 for details).

As an alternative approach to estimate the single mutation effects, we calculated the median isoform frequency across all minigene variants that harbour a given mutation, and compared these numbers to the estimation of the regression model (Supplementary Fig. 4b). If enough minigene variants with the mutation are present in the library, this procedure should average out the effect of accompanying mutations. The median isoform frequency for a mutation was independently calculated for each isoform category and treated as a representative measure of the splicing effect of this particular mutation.

**Estimation of the inference accuracy of the model.** The training data set contained about ~600 mutations that were measured also as single mutations in individual minigenes (Supplementary Table 1). We used these single mutation minigenes to estimate the inference accuracy of the model, and to assess the dependency of the inference accuracy on the occurrence of a mutation in the data set. For each such mutation, the following cross-validation procedure was repeated: The single mutation minigene was removed from the data set before fitting the regression model, and kept for the evaluation of the regression results. The remaining minigenes containing the particular mutation were removed from the data set successively and each time the effect of the mutation was assessed by regression and the estimation compared to the single mutation minigene value. In this way, we obtained estimates for the prediction error based on 1 up to $n-1$ minigenes containing a particular mutation, where $n$ is the total occurrence of the mutation in the data set. In some cases, estimation of mutational effects was not possible from a reduced data set, e.g., the prediction error for a particular mutation was estimated only for occurrences between $m$ and $n-1$, with $1 < m \leq n-1$. Finally, the standard deviation of the inference errors for all mutations was estimated for each measured frequency (Fig. 2c).

**Significant mutation effects and synergistic interactions.** The estimated single mutation effects on splice isoform ratios as obtained by linear regression could be used to predict single mutation effects on each splice isoform frequency ($p_i$) (see Supplementary Note 2 for details). To quantify the effects of each individual mutation on each isoform frequency, we calculated a z-score value from the model-derived single mutation effects, using the mean and standard deviation of the 591 wt minigene variants: $\frac{(p_i^{mutation} - \text{mean}(p_i^{wt}))}{\text{Standard deviation}(p_i^{wt})}$. The z-scores were independently calculated per replicate and later averaged. Only mutations present in all three replicates were kept for further analyses.

In order to combine the evidence from the three replicate experiments, we applied Stouffer's test to combine the z-scores[65]. The resulting standard-normally distributed metric was converted into a $P$ value and subjected to multiple testing correction (Benjamini–Hochberg). We considered a mutation as significant for a given isoform if it displays (i) ≥5% change in isoform frequency compared to the mean of the 591 minigene variants ($\Delta$IF ≥ 5%), and (ii) less than 5% false discovery rate (FDR, adjusted $P$ value <0.05). Combining all six isoform categories, this approach identified 778 and 1022 splicing-effective mutations in HEK293T and MCF7 cells, respectively (Supplementary Table 2). These accumulated into 469 and 550 splicing-effective positions, i.e., nucleotide positions in the *RON* minigene where at least one out of three possible mutations shows a significant effect on at least one isoform.

To calculate z-scores for synergistic interactions between mutations and *HNRNPH* knockdown from the model-derived isoform ratios, we divided the log-transformed fold change in isoform ratios (KD over control condition) by the wt variation (standard deviation; see Supplementary Note 3). z-scores were calculated by replicates and then averaged, removing mutations that were not present in the three replicates under KD conditions. We then used Stouffer's test and multiple testing correction as above. Since the uncertainty of the synergy z-score (measured as the standard deviation between replicates) increases near 0% AE inclusion due to boundary effects (Supplementary Fig. 7g), we excluded the splice sites (positions 209–210, 298–299, 443–444, 523–524 and 689–690) mutations that on their own completely abolish splicing (<1% isoform frequency under control conditions). To identify significant synergistic interactions, we applied a cutoff at 0.1% FDR (adjusted $P$ value <0.001). Additionally, we required a consistent directionality of the synergistic effects in all three replicates. Combining the five different isoform ratios, this approach identified 354 significant synergistic

interactions ($|z\text{-score}| > 2$) on 278 positions between mutations and *HNRNPH* knockdown in MCF7 cells (Supplementary Table 2). Applying more stringent cutoffs at $|z\text{-score}| > 3$ or $>5$ identified 222 or 66 significant synergistic interactions, respectively (Supplementary Table 2).

**Characterisation of splicing-effective positions**. Splice site strengths were predicted using the sequence analysis software MaxEntScan[29] for all mutations in the positions considered by MaxEntScan (278–300 nt and 442–450 nt for the 3′ and 5′ splice site, respectively; Supplementary Fig. 9a, b). PhyloP scores[66] were retrieved from the UCSC Genome Browser (http://genome.ucsc.edu/cgi-bin/hgTables; table: Mammal Cons, PhyloP46wayPlacental) for the genomic coordinates corresponding to the *RON* minigene (chr3:49933134–49933840, human genome version hg19; Supplementary Fig. 9c).

**Annotation of splice-regulatory RBP binding sites (SRBS)**. We used the Scan Sequence tool of the ATtRACT database[34] to identify potential RBP binding sites along the *RON* wt minigene sequence. Duplicated records, e.g., due to overlapping database entries from different experimental methodologies, were removed. We retained only those binding sites for which ≥60% of positions were identified as splicing effective in our screen. This step was independently performed for each splice isoform. Within each RBP, these binding sites were then collapsed if they shared an overlap of ≥2 nt and still harboured ≥60% splicing-effective positions for at least one isoform after collapsing, if they did not fulfil this condition, they were kept unmerged. For the comparison in Supplementary Fig. 12b, the HNRNPH SRBS within each cluster were extended by 2 nt. Nucleotide positions in the two isolated SRBS in the constitutive exons were excluded from this analysis.

In order to connect mutation effects to HNRNPH's sequence specificity, G-run-disrupting mutations were defined as a G-to-H mutation at any position of the G-run (used in Fig. 4c), while the two possible H-to-G mutations in immediately neighbouring positions were counted as G-run-extending. Figure 4b compares the median splicing effect (average of three biological replicates) of all G-run disrupting versus extending mutations for the 22 predicted HNRNPH SRBS.

**Analysis of TCGA and GTEx data**. Normalised gene expression data for 11,688 post mortem samples from 30 human tissues, collected from 714 non-diseased human donors, were retrieved from the GTEx project[36] (v7). Normalised gene expression data from TCGA tumour samples (https://cancergenome.nih.gov/) were retrieved from Firebrowse (http://firebrowse.org/). Alternative splicing for both data sets was quantified using *psichomics* (version 1.2.1, https://github.com/nuno-agostinho/psichomics), using the default minimum coverage to calculate *RON* exon 11 PSI values. We quantified both gene expression and *RON* exon 11 PSIs for 2743 normal samples, from 24 healthy human tissues, and 4514 tumour samples, from 27 cancer types. The comparison of *HNRNPH2* expression between tumours from TCGA (9807 samples) and healthy tissues from GTEx (7851 samples) was done using TPM values calculated at Toil[67], which are already normalised for comparison.

**Calculation of single mutation effects in cancer**. Exome sequencing data from TCGA tumour samples were downloaded from Genomic Data Commons Data Portal (https://portal.gdc.cancer.gov/). We identified a total of 153 patients bearing 55 different mutations within the region of our *RON* minigene (Supplementary Data 5). The impact on splicing of each mutation in the TCGA tumour samples was quantified, per cohort, as the difference of *RON* exon 11 skipping (calculated as $1 - \text{PSI}$) between mutated and non-mutated tumour samples. These differences were correlated with those derived from the skipping isoform frequencies observed in our screen for each mutation. Since we observed that the correlation was affected by the minimum read coverage used to calculate PSIs, we restricted the correlation analysis to cohorts with an average of more than 24 reads mapping to the involved splice junctions (resulting in 117 patients from 14 cohorts harbouring 36 different mutations; Fig. 3c). The intrinsic variability of *RON* exon 11 inclusion levels in TCGA patient samples was calculated as the standard deviation of *RON* exon 11 PSI in unmutated TCGA tumour samples (i.e., without a given mutation) from cohorts considered in Fig. 3c and with more than 24 reads mapping to the involved splice junctions.

**Identification of candidate RBPs**. A recent large-scale RBP KD screen tested the KD effect of >200 RBPs on splicing of *RON* exon 11 and other alternative exons in HeLa cells[35]. The study used *z*-scores calculated from the PSI upon siRNA treatment and the median absolute PSI deviation, divided by its standard deviation. A positive *z*-score indicates more AE inclusion upon RBP KD. Using a cutoff of $|z\text{-score}| > 1.5$, 125 RBPs showed a substantial effect on *RON* exon 11 splicing. These include 17 RBPs that also have predicted SRBS in the *RON* minigene.

In order to identify potential regulators of *RON* exon 11 splicing in humans, we searched for RBPs whose expression correlated with *RON* exon 11 splicing in cancer. The correlation analysis was performed with 190 pre-selected RBPs, consisting of 65 identified via ATtRACT, 108 identified in the previously published RBP KD screen[35] and 17 common to both approaches. The mRNA expression levels of the RBPs were Spearman-correlated with *RON* exon 11 inclusion levels across TCGA tumour samples (Supplementary Data 7 and Supplementary Table 3). The significance of those correlations (ranked by minus base-10

logarithm of the associated *P* value) was tested against those of all RBPs retrieved from[8] and of all protein-coding genes using Gene Set Enrichment Analysis (GSEA) tool[68,69]. RBPs and protein-coding genes were first restricted to the ones showing at least the same average expression value as the least expressed pre-selected RBP, known to be highly expressed in cancer, so that GSEA was not biased by gene expression ranges. Moreover, we performed linear regressions between the expression of each of the 190 pre-selected RBPs and *RON* exon 11 PSI in TCGA tumour samples, using the resulting slopes to quantitatively assess the relative magnitude of association between each RBP and *RON* exon 11 splicing.

**Analysis of cooperativity and switch-like splicing behaviour**. Changes in percent spliced-in (ΔPSI) data for *RON* exon 11 inclusion from the endogenous *RON* gene and the wt *RON* minigene measured at different *HNRNPH* knockdown (KD) and overexpression (OE) levels (Supplementary Fig. 15a–d, 16) were fitted using the Hill function

$$y(x) = y_{\max} - \frac{(y_{\max} - y_{\min})x^{n_H}}{x^{n_H} + \text{EC50}^{n_H}},$$

with $x$ and $y$ being vectors of experimentally determined HNRNPH levels and corresponding splicing outcomes (ΔPSI), respectively (Fig. 6b). $y_{\min}$, $y_{\max}$, EC50, and $n_H$ are fitted parameters. Fitting was done by minimising the residual cost function

$$\chi^2 = (\Delta\text{PSI} - y(\text{HNRNPH}))/\sigma_{\Delta\text{PSI}},$$

where $\sigma_{\Delta\text{PSI}}$ denotes the standard deviation of the PSI measurement. Minimisation was done using the Matlab nonlinear least-squares solver *lsqnonlin*. The parameter ranges used during fitting were $y_{\min} \in [-0.5,0]$, $y_{\max} \in [0,0.5]$, EC50 $\in [0.1,2]$ and $n_H \in [1,20]$. The optimal parameter values found were

1. for the endogenous *RON* gene: $y_{\min} = -0.11$, $y_{\max} = 0.36$, EC50 $= 0.93$, $n_H = 17.4$
2. for the wt *RON* minigene: $y_{\min} = -0.11$, $y_{\max} = 0.3$, EC50 $= 0.94$, $n_H = 13.8$

Confidence intervals were determined for all parameters by using a profile likelihood approach. For each fitted parameter $\theta$, the following workflow was repeated: The parameter was assigned successively a number of values around its optimal value $\theta_0$ listed above. While keeping this parameter at the fixed value, the remaining parameters were optimised and the value of the corresponding cost function was determined. Thus, the dependence of the cost function $\chi^2(\theta)$ on the parameter value around the minimum corresponding to the optimal value $\theta_0$ was determined. The likelihood-based confidence interval for this parameter is defined by

$$[\theta, \chi^2(\theta) - \chi^2(\theta_0) < \chi^2(\alpha, 1)],$$

where $\alpha$ is the confidence level and $\chi^2(\alpha,1)$ is the $\chi^2$ distribution with degree of freedom 1. For each parameter, the 95% confidence intervals were found by determining the values $\theta$ on both sides of $\theta_0$, for which the likelihood $\chi^2(\theta)$ crosses the threshold $\chi^2(\theta_0) + \chi^2(0.95,1)$.

The 95% confidence intervals found for the endogenous *RON* gene were:

$$y_{\min} \in [-0.12, -0.1], y_{\max} \in [0.28, 0.43],$$
$$\text{EC50} \in [0.89, 0.95], n_H \in [10.8, 35.2],$$

and for the wt *RON* minigene:

$$y_{\min} \in [-0.14, -0.08], y_{\max} \in [0.3, 0.31],$$
$$\text{EC50} \in [0.93, 0.95], n_H \in [10.4, 17.7],$$

**Code availability**. Code that was used to generate the presented data is available from the corresponding authors upon request.

**Data availability**. The sequencing data generated in this study are available from ArrayExpress under the accession numbers E-MTAB-6216 and E-MTAB-6217 (RNA-seq), E-MTAB-6219 (DNA-seq), E-MTAB-6220 and E-MTAB-6221 (iCLIP). All other data supporting the findings of this study are available from the corresponding authors on reasonable request.

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

## Acknowledgements

The authors would like to thank the members of all participating labs for their support and discussion. We gratefully acknowledge the Institute of Molecular Biology Core Facilities for their support, especially the Genomics and the Bioinformatics Core Facilities, and the use of the Illumina NextSeq 500 instrument (INST 47/870-1 FUGG) as well as Teresa Maia (NMorais Lab, iMM) for assistance with TCGA data retrieval and analyses and Tina Han for help and technical support. We would like to thank Guiseppe Biamonti and Heiner Schaal for advice on the *RON* minigene. The results published here are in part based upon data generated by TCGA managed by the NCI and NHGRI. Information about TCGA can be found at http://cancergenome.nih.gov. This work was funded by a joint DFG grant (ZA 881/2-1 to K.Z., KO 4566/4-1 to J.K. and LE 3473/2-1 to S.L.). K.Z. was also supported by the LOEWE program Ubiquitin Networks (Ub-Net) of the State of Hesse (Germany) and the Deutsche Forschungsgemeinschaft (SFB902 B13). N. Barbosa-Morais' laboratory is supported by EMBO (Installation Grant 3057) and Fundação para a Ciência e a Tecnologia, Portugal (FCT Investigator Starting Grant IF/00595/2014). S.L. acknowledges support by the German Federal Ministry of Research (BMBF; e:bio junior group program, FKZ: 0316196). The Institute of Molecular Biology (IMB) gGmbH is funded by the Boehringer Ingelheim Foundation.

## Author contributions

S.B. established the high-throughput screening approach and performed most experiments. S.T.S. performed most bioinformatics analyses. M.E. and S.L. designed the mathematical modelling approach and performed the analyses. M.C.-L. annotated putative RBP binding sites and analysed mutation effects and synergistic interactions. M. S. contributed to the RNA sequence annotation. B.P.d.A. and N.L.B.-M. performed the analyses of the TCGA and GTEx data sets. F.X.R.S. performed iCLIP experiments, and L. S. did validation experiments. A.B. performed iCLIP and RNA-seq data processing as well as splice isoform quantification. S.E. and K.Z. supervised the bioinformatics analyses. J.K. conceived the project and supervised the experimental work. S.B., S.L., J.K. and K.Z. wrote the manuscript with help and comments from all co-authors.

## Additional information

**Competing interests:** The authors declare no competing interests.

