## [Peer Review File · Nature Communications]

Reviewers' comments:

Reviewer #1 (Remarks to the Author):

The manuscript describes a high-throughput screening approach to define the cis-regulatory network regulating the alternative splicing (AS) of Ron exon 11.

Ron encodes for a tyrosine kinase receptor involved in several biological processes such as cell dissociation, migration and invasion. Several Alternative Splicing (AS) isoforms of Ron were found expressed in cancers. Among these is RonD165, generated through skipping of exon 11, increased in human cancers and involved in EMT and invasive tumorigenesis. RonD165 is a model gene that has been used to understand how alteration in expression levels (or activity) of specific RNA-binding proteins (RBPs) could contribute to tumor progression.

In this manuscript, the authors used systems approaches combined with mathematical modelling to investigate the complex regulation of Ron exon 11 AS and to relate Ron mutations to splicing outcomes in cancer patients.

Specific points:

1) Several RBPs were reported to regulate Ron exon 11 splicing. Here, the authors described hnRNP H as a master regulator of RonD165 production that acts by binding multiple cis-regulatory elements in a cooperative manner. Nevertheless, the notion that hnRNP H promotes skipping of Ron exon 11 is not novel (Lefave CV et al EMBO J. 2011). Cooperative hnRNP H binding to multiple Ron motifs was tested by performing a hnRNP H knockdown titration in which MCF7 cells were transfected with increasing amounts of hnRNP H-specific siRNA and splicing of the Ron exon 11 (from the minigene as well as the endogenous Ron gene) was measured. It would be important to show opposite results by hnRNP H gain of function by performing overexpression experiments with increasing amounts of hnRNP H.

2) Related to the experiments shown in Figure S12 in some case there is no correlation between down-regulation of hnRNP H and Ron exon 11 splicing. For example: in the second western blotting of the panel B (second lane) performed to determine the hnRNP H knockdown, the quantification of hnRNP H expression levels is 114 (more than the value shown in the first lane that is the Ctr, non targeting control siRNA). Despite there is no depletion of hnRNP H, the authors show that there is a drastic inclusion of the endogenous Ron exon 11 (panel A on the left). The same for a number of lanes of the third western blotting (the lower) in which changes in the expression levels of hnRNP H do not always correlate to the endogenous Ron exon 11 splicing. The authors must address these discrepancies.

3) The production of RonD165 was associated to activation of EMT, a process involved in tumor progression and metastasis formation of human epithelial cancers. To address the functional relevance of the regulation of Ron splicing by hnRNP H authors must evaluate if hnRNP H is able to regulate EMT through the splicing of Ron. Do hnRNP H expression levels (by over-expression and knockdown) affect EMT? Is the effect specifically mediated by the production of RonD165 isoform? For example, over-expression of hnRNP H combined with RonD165 specific knockdown should be performed to address this point.

4) In the TCGA dataset a significant negative correlation between hnRNP H expression levels and Ron exon 11 inclusion was observed. Is this also correlated with the metastatic potentials of the tumor samples. It would be important also to correlate the expression of hnRNP H and RonD165 with the survival of cancer patients.

Minor points

Page 3, line 62: the acronym RBPs must be introduced, whereas it is reported below (line 80).

Reviewer #2 (Remarks to the Author):

This manuscript by Braun and colleagues presents a well-designed and beautifully crafted approach to identify mutations that disrupt splicing regulation. It uses a variety of expertise including state of the art genomics, molecular biology, biochemistry, bioinformatics and mathematical modeling. The results are in general superbly controlled and the data is presented with useful illustrations. The approach is applied to dissect the alternative splicing unit of RON exon 11. Because on average 3.8 mutations is produced per minigene, mathematical modeling is used to deconvolute these effects and attribute the impact of 1800 individual mutations. Sequence information is then used to predict the identity of trans-acting regulators whose recruitment may be affected by the mutations. One of the most important regulator identified by this approach turns out to be hnRNP H. hnRNP H is already known to be a potent regulator of RON splicing, and the most potent mutations in exon 11 that affect its splicing occur in a region that has already been linked to hnRNP H (LeFave et al. EMBO J. 2011). The more original contribution here lies in building a case for cooperative binding for hnRNP H as the basis for regulation.

I find the overall approach very interesting and of broad interest, and therefore worthy of reporting. However, novelty of contribution could be improved by the analysis of more mutations to better assess the cooperativity of hnRNP H binding to exon 11. Additional concerns should also be addressed as indicated below.

1. Not all active mutations are expected to disrupt cis-acting elements that are bound by proteins. For example, a mutation can disrupt or create a secondary structure. Also, the authors have themselves identified mutations that create better binding sites for hnRNP H, and this could happen for dozens of RBPs. This caveat should be indicated somewhere. This point affects their conclusion that the RON alternative spliced region is densely packed with regulatory elements, something we already know for FAS exon 6, SMN exon 7, CFTR exon 12 but which requires systematically mutating each position.

2. 778 mutations significantly alter RON alternative splicing with a Δ PSI >5%. Some discussion of the range of amplitudes would be warranted. The impact of most mutations appears to be minor (between 5-10%). How many mutations give Δ PSI larger than 10%, larger than 25%? Do the best shifting mutations tend to be neighbors? hnRNP H binding sites?

3. I have some reservation with the epistasis analysis. The authors propose that if a mutation causes a reduced response to hnRNP H knockdown (relative to wt), this should be taken as an indication that hnRNP H is involved. However, take the following case: inclusion for the wt goes from 20% to 100% with the H knockdown (Δ PSI of 80%), whereas a mutation that causes a 60% inclusion moves to 100% with the H knockdown (here a Δ PSI of 40%, hence a shift smaller than wt). My point is that another regulator could be affected by the mutation and because the mutation yields by itself a higher PSI and that maximal inclusion is reached upon KD, the response to H depletion is reduced but not necessarily because the mutation affects hnRNP H. The fact that 707 mutations (out of 1787) show epistatic interactions with hnRNP H suggest that something is off and that the premise to assume epistasis in all cases may be wrong. Nevertheless, I agree that some known hnRNP H binding sites that have an impact on splicing when mutated in exon 11 seem to display epistatic interactions.

4. Cooperative binding analysis is tantalizing but needs to be better documented by testing other mutations that affect putative H binding sites. As binding of H to G348 does not appear to be affected by G305A, testing G348C should also be tested, as well as other weaker hnRNP H sites and sites that do not respond to KD in exon 11.

5. I could not find Supplementary Data 2.

Reviewer #3 (Remarks to the Author):

Summary:

In this work Braun et al perform extensive mutagenesis screen of RON exon 11 and build a linear model for the effect of the identified mutations on splicing outcome. They measure the accuracy of their model using train/test and comparing it to a strawman which is simply the median effect of a given single point mutation across all mutation combinations in which it was measured. They identify several SF which could be regulating exon 11 using several lines of previous evidence (SF KD, SF CLIP peaks), and correlate exon 11 inclusion to the expression of those in TCGA and GTEX. The most prominent regulator they identify and focus on is hnRNP H, which was previously reported and studies in this context (Ghigna, et al "Cell motility is controlled by SF2/ASF through alternative splicing of the Ron protooncogene". Mol Cell 2005). They KD hnRNP-H and compare the effect of their library of mutations in WT and hnRNP-H KD. They show different hnRNP-H sites have antagonistic and non-linear effects, and its expression levels correlate with exon11 inclusion in TCGA/GTEX. Finally, they show their model predictions for mutation effect on splicing correlate with the changes observed in TCGA cancer patients who harbor these mutations.

The authors should be congratulated for creating such an extensive work that spans many experiments, modeling, and large-scale genomic analysis all tied nicely together. The writing is clear, many of the figures are beautiful and smart. We really liked how the authors went to real data (TCGA/GTEX) to test their findings, compared the effect of the mutations in WT to hnRNP-H KD, and

identify complex interactions for hnRNP-H sites. Of note, such analysis was not done in previous highly related work (which is also good) by Julien et al and Rosenberg et al. With that mind, we still found many issues which should be addressed in full.

Major Comments:

We struggled with some of the basic definitions and choices of the predictive model. Some of those were confusingly described, some seem like they might be misguided. Specifically:

The authors repeatedly emphasize that they do not model PSI directly/linearly but use ratios instead and that this allows them to avoid issues with an otherwise linear model (paragraph from line 160). Instead, they explain in the supp that they model ratios from a baseline, which yields 5 *separate* regression models. But, as they acknowledge, this gives them a free parameter so they renormalize to get PSI values. So, after all the nice explanations of kinetics (which could serve as a motivation), the math as far as we can tell boils down to be equivalent to what you would get from a softmax regression. (eq. 10,11 in supp). Moreover, they use log PSI, so the natural derivation given what they are trying to model and what they eventually output is a (sparse) logistic regression model using a softmax, and optimizing the conditional log-likelihood, or equivalently the cross entropy. But the authors instead keep those as separate models, optimize each using an L2 distance, and then renormalize. This seems to get them into trouble with what they refer to as nonlinear effects (page 4 bottom in supp) as the info between the separate optimizations is not shared. Running such multiclass (sparse) logistic regression with cross entropy, L1, or L2 should be a straightforward exercise using packages such as scikit-learn. The authors should definitely try those and if we missed something or these are inferior they should clearly explain/demonstrate this.

The issue about what is linear or not and what they compare to affects their claims/statements in the discussion as well regarding what the model is doing (lines 428-433). It will also affect their discussion of linearity and related work in lines 434-445. Specifically when they state the “majority” is not linear (line 436) it’s not clear what is that “majority” and again, what is the definition of this linearity. If they took out the fraction of close proximity mutations done in Julien et al 2016 would their results agree, or is it also a result of how they defined “linearity”?? Also note that Rosenberg et al used a quite different model over k-mers, something not mentioned.

* The authors exclude ~6% of the variants as they can not model those, even though they acknowledge these can have a significant effect and create many variants (line 141, page 5 top in supp). Since they are interested in modeling cancer-specific variations it might very well be that those 6% are particularly important/deleterious. What are the fractions of those unmodeled in the cancer data? Do they have a significant effect based on their metric even without a model and correlation to what happens in cancer (e.g. median over all variants including those?). Related to that, the model can not handle the 608 short insertions and deletions (line 684) which are ~3% of the variants they created, but this model limitation is not discussed in the main text.

* The fact they don’t find increased conservation of regulatory elements in the introns (line 235) is

not well explained/discussed. This is also in clear contrast to many other works that show highly conserved intronic regulatory elements for splicing. One possible explanation would be that the intronic positions they identified exhibit a smaller effect (do they? You could check!) and/or that the previous works focused more on highly conserved tissue-specific regulation while in this case there are just general (subtle?) effects on the base level of inclusion. Either way, authors should extend the discussion here.

* The results on cooperative hnRNP H sites leading to a splicing switch are very nice. The authors gloss over whether their mutation model is able to capture that or not, which should be noted. Also, this result should be noted as inline with previous experimental results and splicing code predictions made for CU/CUG elements around daam1 exon 16 (Barash et al Nat 2010).

* We found the use of the term “epistatic” interactions between hnRNP H KD and mutations to be extremely confusing. In general, the term is commonly used for the non-additive effect of genetic variations, but not here. Furthermore, the epistatic interaction here happens when a mutation in WT behaves differently than under hnRNP H KD. So “normal” interaction i.e. when it’s a BS for hnRNP H, are now denoted “epistatic”! Worst, the term is used differently in Julien et al for variant combinations, which makes it all the more confusing. Why??

* The authors gloss over the fact that their results show “epistatic” effect for hnRNP-H for 468/550 positions (!!) in MCF7 cells. This is worth explaining/discussing.

Figure specific comments:

=====

Fig2B:

This figure you can't see much. Could plot it separately for each variant at least in the supp. They could also plot for each variant the max deviation in PSI so we see how good/bad are the variants prediction e.g. X% of them vary more than Y% PSI than the observed.

Fig2C:

This is a nice result and analysis. But it would be good to know (a) what is the distribution of #occurrences i.e. the PDF/CDF of mutation occurrences in the data (the x-axis here) (b) combining the above and their current graph, compare to a median (strawman) how many (CDF) the improve by more than X% in error compared to the strawman. This would give a sense of what is the overall gain in accuracy from the model compare to only using the experimental data.

Fig3C:

This result is somewhat disappointing, and its presentation both here and in the main text is misleading. In practice, the r correlation value hinges on two clear outliers which have a strong effect, G370T and G297A. They should report the correlation without these as well and discuss this. Also, G297A is a splice site one so not very interesting. The G370T is more interesting, but is it in an hnRNP H site? something else? it is not even shown in Fig B for some reason.

To show the utility of their model they should compute and report the same correlation if they used just the median value for each of those mutations. This will show the relative gain of their model compared to using the experimental results directly.

Fig3E:

This is a nice connection but it is not fully explored. The pvalues on the right are somewhat misleading as they relate to the effect measured of KD in a completely different work. What would be much more informative would be to see the correspondence between their model prediction for taking away the sites of a SF and the KD experiment effect. Of course, a site could be shared, nonetheless, it would indicate how well they can perform such an analysis with their model.

Minor Comments:

=====

In various places, results/conclusions are overstated. For example:

The title is somewhat misleading as it implies decoding of the entire RON gene, in practice only for exon 11.

The abstract overstates the results, should be toned down. We don't see the results, as presented, offering "insights on the functional impact in human disease".

Similarly, at the end of the intro, the authors state: "Our approach promises new insights into the molecular function of cancer-associated mutations and the mechanisms of alternative splicing regulation in general." Again, this seems like an overstatement given what is actually delivered, which only applies (with limited capacity, as discussed above) to exon 11 in RON.

In line 421 in the discussion, authors summarize that their model "can reliably predict the 421 individual effects of almost 1,800 mutations". This statement is misleading: The model is not able to predict the effect of *any* mutation it has never seen. It can infer the effect of a single mutation given several combinations in a manner which is more accurate than trying to simply guess by the mean/median.

In lines 475-478 the authors claim their method can be extended further enable to reconstruct complete splicing code. Given the usage of tailored mini-gene per exon/gene and KD of each SF for each of those, it seems this claim for large-scale analysis should be toned down or removed.

Line 480 states the study shows hnRNP-H is a "master" regulator. Why? "Key" or "important" seems more appropriate here.

* References choices in the intro seem to reflect some knowledge of recent literature but are not aligned well with the statements and background. For example, Yang et al 2016 is relevant for showing the effect on protein-protein interaction, but this was shown well before that on a large scale in Ellis et al Mol Cell 2012. The observation that over 90% of human genes undergo AS should

be attributed to the back to back papers Pan et al Nat Gen 2008 and Wang et al Nat 2008. Similarly, the term 'splicing code' is usually attributed to the Wang and Burge RNA 2008 review that defined what is such a code and specified it as a long-term goal, or to Barash et. al Nat 2010 which offered the first such code derivation.

* Just as the references seem lacking, the authors do not draw connections to numerous previous works. For example:

In line 214 they report that they were surprised to find the splice site of downstream exon12 affects the skipping of exon 11. The strength of the up/down sites were already included in the original splicing code as an informative feature (Barash et al, Nat 2010) and numerous other works identified binding SF around those as affecting ES (e.g. Llorian et al NSMB 2011) so the result should not be such a surprise.

Line 486-489 again this conclusion is in line with numerous works, from classical SrcN1 works by Doug Black mapping it's regulatory elements in a series of papers, to more computational works as in Barash et. al Nat 2010.

By far the most relevant work is Julien et al. The authors do mention it several times, but given the strong connection it would help readers to more directly compare/contrast the various conclusions/results for each section. This should not be seen as taking away from the impact/relevant of this paper as there is a lot of unique results in this paper already (see opening statement)

Response to Reviewers' Comments:

We would like to thank the Reviewers for their very positive feedback and their constructive criticism. We addressed all comments which substantially improved the manuscript. Please find our detailed responses below.

Reviewers' comments:

Reviewer #1 (Remarks to the Author):

The manuscript describes a high-throughput screening approach to define the cis-regulatory network regulating the alternative splicing (AS) of Ron exon 11.

Ron encodes for a tyrosine kinase receptor involved in several biological processes such as cell dissociation, migration and invasion. Several Alternative Splicing (AS) isoforms of Ron were found expressed in cancers. Among these is RonD165, generated through skipping of exon 11, increased in human cancers and involved in EMT and invasive tumorigenesis. RonD165 is a model gene that has been used to understand how alteration in expression levels (or activity) of specific RNA-binding proteins (RBPs) could contribute to tumor progression.

In this manuscript, the authors used systems approaches combined with mathematical modelling to investigate the complex regulation of Ron exon 11 AS and to relate Ron mutations to splicing outcomes in cancer patients.

Specific points:

1) Several RBPs were reported to regulate Ron exon 11 splicing. Here, the authors described hnRNP H as a master regulator of RonD165 production that acts by binding multiple cis-regulatory elements in a cooperative manner. Nevertheless, the notion that hnRNP H promotes skipping of Ron exon 11 is not novel (Lefave CV et al EMBO J. 2011). Cooperative hnRNP H binding to multiple Ron motifs was tested by performing a hnRNP H knockdown titration in which MCF7 cells were transfected with increasing amounts of hnRNP H-specific siRNA and splicing of the Ron exon 11 (from the minigene as well as the endogenous Ron gene) was measured. It would be important to show opposite results by hnRNP H gain of function by performing overexpression experiments with increasing amounts of hnRNP H.

We agree with the Reviewer that overexpression experiments are a valuable complement to the gradual *HNRNPH* knockdown shown in the original Fig. 6b. We therefore performed additional experiments in which we overexpressed HNRNPH1 at increasing levels in MCF7 cells. Together with the endogenous protein, this resulted in a 197% and 218% overexpression. Semi-quantitative RT-PCR showed a reduction in *RON* exon 11 inclusion for the minigene as well as the endogenous *RON* gene. Fitting the Hill equation to the complete dose-response curves involving knockdown and overexpression data yielded high Hill coefficients ($nH = 17.4$, confidence interval (CI) [10.8,35.2] for endogenous *RON*; $nH = 13.8$, CI [10.4,17.7] for *RON* minigene). Thus, the uncertainty in the Hill coefficients could be reduced by the additional overexpression experiments, further supporting the initial conclusions concerning cooperative regulation of splicing by HNRNPH.

The new data are now shown in **Figure 6b** and **Supplementary Figure 15b,d**.

2) Related to the experiments shown in Figure S12 in some case there is no correlation between down-regulation of hnRNP H and Ron exon 11 splicing. For example: in the second western blotting of the panel B (second lane) performed to determine the hnRNP H knockdown, the quantification of hnRNP H expression levels is 114 (more than the value shown in the first lane that is the Ctr, non targeting control siRNA). Despite there is no depletion of hnRNP H, the authors show that there is a drastic inclusion of the endogenous Ron exon 11 (panel A on the left). The same for a number of lanes of the third western blotting (the lower) in which changes in the expression levels of hnRNP H do not always correlate to the endogenous Ron exon 11 splicing. The authors must address these discrepancies.

We believe that the moderate variation in the protein quantification as pointed out by the Reviewer reflects the inherent variability in experimental measurements which always underlie a certain biological and experimental noise. As a result, the interpretation of individual data points can be misleading. By using replicates, we overcome the problem to discern true differences in HNRNPH levels and splicing changes from random variation. For each data point, we had therefore performed three independent repetitions using independently grown cell cultures. We would like to emphasise that our conclusions, i.e. that small changes in HNRNPH levels induce a switch-like splicing response, are exclusively drawn from the measured averages.

In order to highlight the variability between replicates, we added information on mean and standard deviation for concentration levels in the gradual *HNRNPH* knockdown and overexpression in the revised **Supplementary Figure 15**.

3) The production of RonD165 was associated to activation of EMT, a process involved in tumor progression and metastasis formation of human epithelial cancers. To address the functional relevance of the regulation of Ron splicing by hnRNP H authors must evaluate if hnRNP H is able to regulate EMT through the splicing of Ron.

Do hnRNP H expression levels (by over-expression and knockdown) affect EMT? Is the effect specifically mediated by the production of RonD165 isoform? For example, over-expression of hnRNP H combined with RonD165 specific knockdown should be performed to address this point.

This question had already been addressed in a previous work of Lefave et al., *EMBO J*, 2011.

In their study, the authors employed a 'splicing switchback' experiment (presented in Figure 6 of their paper): First, they used antisense morpholinos against both splice sites of exon 4 in *HNRNPH1* to induce skipping of this internal exon. The resulting frameshift introduces a premature stop codon, which efficiently targets the *HNRNPH1* transcripts into nonsense-mediated decay, leading to a significant HNRNPH downregulation at the RNA and protein level. In a second step, this HNRNPH depletion was combined with a second set of antisense morpholinos (H4) to induce skipping of *RON* exon 11 (splicing redirection). RT-PCR confirmed that hnRNP H depletion led to nearly 100% inclusion of *RON* exon 11, which was reverted to control levels when combined with H4 antisense morpholinos against *RON* exon 11 (Figure 6C, top). A matrigel invasion assay showed that the hnRNP H depletion strongly reduced the invading capability (tested in T98G glioma cells and HeLa cells). Importantly, the

simultaneous addition of the H4 antisense morpholinos and the resulting induction in *RON* exon 11 skipping was sufficient to increase cell motility and thus partially rescue the phenotype, even in the absence of HNRNPH. The authors concluded that “hnRNPH levels contribute to the invading properties of glioblastoma and other cancer cells, at least in part through modulation of *RON* exon 11 splicing.”

4) In the TCGA dataset a significant negative correlation between hnRNP H expression levels and *Ron* exon 11 inclusion was observed. Is this also correlated with the metastatic potentials of the tumor samples. It would be important also to correlate the expression of hnRNP H and *RonD165* with the survival of cancer patients.

We agree with the Reviewer that the metastatic potential of tumours and patient survival are important parameters and therefore performed the suggested analyses for further investigation.

Consistent with a role of constitutively active *RON* Δ 165 in cancer, we showed in the original version of the manuscript that *RON* exon 11 inclusion levels are commonly reduced in TCGA tumours (mean PSI 67%) compared to healthy tissues from GTEx (mean PSI 76%, P -value < $2.2e-16$; Mann-Whitney-Wilcoxon test). To similarly support a potential role of HNRNPH, we now extend this analysis to show that *HNRNPH2* consistently shows higher expression in the tumours compared to healthy tissues (mean TPM 57.88 vs. 46.29; P -value < $2.2e-16$; Mann-Whitney-Wilcoxon test). Even though the observed changes in *HNRNPH2* are rather small, our experimental results suggest that this can have drastic impact on *RON* exon 11 (and potentially other target exons). These findings support the notion that HNRNPH-mediated repression and *RON* exon 11 skipping are associated with human cancer.

We report these results in the revised manuscript.

Following the Reviewer’s suggestion, we tested the association between *HNRNPH2* expression or *RON* exon 11 inclusion levels and tumour stage IV (as a surrogate of association with metastatic potential) across TCGA cohorts with stage IV samples. Requiring a minimum of 10 samples per cohort, *HNRNPH2* expression was tested in 20 cohorts and *RON* exon 11 inclusion was tested in 12 cohorts. However, we could not find evidence for either *HNRNPH2* or *RON* exon 11 being differently expressed/included between stage IV and any of the other stages (stages I-III) (see **Figure I e** below). A possible explanation for this could be that *HNRNPH* expression and *RON* splicing are not the only determinants of cancer malignancy but contribute, in certain tumours and patients, to tumour progression.

We further tested the prognostic value of *HNRNPH2* expression and *RON* exon 11 inclusion on the survival of cancer patients across TCGA tumour cohorts. Using the R package *survival* (Pearce et al., bioRxiv, doi: <https://doi.org/10.1101/208660>; default parameters) to stratify TCGA patients based on an optimal Kaplan-Meier cutoff, we could not detect a significant association of either measure with patient survival across TCGA cohorts (see **Figure I c,d** below). The exceptions were two cohorts, in which lower expression of *HNRNPH2* was significantly associated with worse prognosis (Adrenocortical Carcinoma and Lower Grade Glioma; **Figure I a,b** below). However, both cohorts showed hardly any expression of *RON* (mean TPM 0.40 and 0.66 for Adrenocortical Carcinoma and Lower Grade Glioma, respectively), suggesting that other HNRNPH-regulated events rather than *RON* exon 11 skipping must be involved in the observed effect. Since our manuscript studies HNRNPH-

mediated regulation in the context of RON splicing, we decided not to include these results into the revised version.

Figure I. Analysis of tumour stage and patient survival.

Minor points

Page 3, line 62: the acronym RBPs must be introduced, whereas it is reported below (line 80).

We corrected this mistake.

Reviewer #2 (Remarks to the Author):

This manuscript by Braun and colleagues presents a well-designed and beautifully crafted approach to identify mutations that disrupt splicing regulation. It uses a variety of expertise including state of the art genomics, molecular biology, biochemistry, bioinformatics and mathematical modeling. The results are in general superbly controlled and the data is presented with useful illustrations. The approach is applied to dissect the alternative splicing unit of RON exon 11. Because on average 3.8 mutations is produced per minigene, mathematical modeling is used to deconvolute these effects and attribute the impact of 1800 individual mutations. Sequence information is then used to predict the identity of trans-acting regulators whose recruitment may be affected by the mutations. One of the most important regulator identified by this approach turns out to be hnRNP H. hnRNP H is already known to be a potent regulator of RON splicing, and the most potent mutations in exon 11 that affect its splicing occur in a region that has already been linked to hnRNP H (LeFave et al. EMBO J. 2011). The more original contribution here lies in building a case for cooperative binding for hnRNP H as the basis for regulation.

I find the overall approach very interesting and of broad interest, and therefore worthy of reporting. However, novelty of contribution could be improved by the analysis of more mutations to better assess the cooperativity of hnRNP H binding to exon 11. Additional concerns should also be addressed as indicated below.

1. Not all active mutations are expected to disrupt cis-acting elements that are bound by proteins. For example, a mutation can disrupt or create a secondary structure. Also, the authors have themselves identified mutations that create better binding sites for hnRNP H, and this could happen for dozens of RBPs. This caveat should be indicated somewhere. This point affects their conclusion that the RON alternative spliced region is densely packed with regulatory elements, something we already know for FAS exon 6, SMN exon 7, CFTR exon 12 but which requires systematically mutating each position.

The Reviewer is correct that not all splicing-effective mutations necessarily mark an RBP binding site. The introduced mutations may indeed affect RNA secondary structure elements that are involved in splicing regulation and could thus be considered as *cis*-acting elements. Such structures can directly impact on splicing e.g. by modulating the accessibility of splice sites and other regulatory sites or via RBP recruitment. In fact, several RBPs were previously shown to interact with specific RNA secondary structures to regulate alternative pre-mRNA splicing. Consistently, we speculated in the discussion that RNA secondary structures might contribute to the cooperative regulation of HNRNPH.

We also agree with the Reviewer that splicing-effective mutations can not only disrupt but also introduce new *cis*-regulatory elements, thereby further increasing the complexity of *RON* splicing regulation. However, we think that most of these mutations will not generate a completely new site. Instead, it appears more likely that they will reinforce a previously existing but weak *cis*-regulatory element, e.g. by shifting a binding site towards a higher RBP binding affinity. Moreover, even though these mutation-generated *cis*-regulatory elements may not have a major impact under normal conditions, they can gain pathophysiological

relevance in cancer when mutations accumulate. We therefore think that it is important to include such mutations in our analysis.

Following the Reviewer's request, we now point to these considerations at several positions in the revised manuscript. To emphasise a possible involvement of RNA secondary structures in splicing regulation, we added to the introduction that both primary sequence as well as secondary structure can form *cis*-regulatory elements important for splicing regulation. In addition, we mention the impact of mutation-generated *cis*-regulatory elements in the Results section and later when discussing the dense regulatory landscape of *RON* exon 11.

2. 778 mutations significantly alter *RON* alternative splicing with a Δ PSI >5%. Some discussion of the range of amplitudes would be warranted. The impact of most mutations appears to be minor (between 5-10%). How many mutations give Δ PSI larger than 10%, larger than 25%? Do the best shifting mutations tend to be neighbors? hnRNP H binding sites?

Indeed, we find that mutations can strongly differ in their effect sizes, with many displaying between 5-10% change. Importantly, however, 47% (HEK293T) and 51% (MCF7) of the splicing-effective mutations trigger changes of >10% in at least one splice isoform (363 out of 778 in HEK293T, and 521 out of 1022 in MCF7), and 1/5 of them exceed 20% (136 and 189, respectively). We thus detect a large number of potent mutations that substantially change the splicing outcome for *RON* exon 11.

In order to visualise the occurrence and spatial distribution of weak and strong mutation effects, we modified the landscapes of splicing-effective mutations in the revised **Supplementary Fig. 8** (previously Figure S6). The plots now overlay the three different thresholds. Additionally, we extended **Supplementary Table 4a,b** to specify the number of splicing-effective mutations and positions in different transcript regions based on the three thresholds (>5%, >10% and >20%).

Overall, these analyses show that the strongest mutations cluster in and around the alternative exon (see **Supplementary Fig. 8**). As expected, hotspots include the splice sites, but also numerous other locations including the HNRNPH SRBS in the alternative exon. Furthermore, we investigated the spacing of significant mutations with >5%, >10% and >20% effects on any isoform frequency. We find that the stronger mutations tend to be neighbours, supporting the notion that they point to *cis*-regulatory elements (see **Figure II** below). Due to space restrictions, we decided to not include the distance analysis in the revised manuscript.

Figure II: Distance to nearest splicing-effective mutation for different isoforms and cutoffs.

3. I have some reservation with the epistasis analysis. The authors propose that if a mutation causes a reduced response to hnRNP H knockdown (relative to wt), this should be taken as an indication that hnRNP H is involved. However, take the following case: inclusion for the wt goes from 20% to 100% with the H knockdown (Δ PSI of 80%), whereas a mutation that causes a 60% inclusion moves to 100% with the H knockdown (here a Δ PSI of 40%, hence a shift smaller than wt). My point is that another regulator could be affected by the mutation and because the mutation yields by itself a higher PSI and that maximal inclusion is reached upon KD, the response to H depletion is reduced but not necessarily because the mutation affects hnRNP H.

In response to the Reviewer's concern and major comment 6 of Reviewer #3 (see below), we now refer to the epistatic effects as 'synergistic interactions'. Please note that this terminology is already used throughout the following paragraph.

We agree with the Reviewer that the analysis of synergy at the level of 'percent spliced-in' (or related metrics) is problematic, as PSI is by definition bounded between 0 and 100%, thereby potentially giving rise to apparent synergy of unrelated perturbations. In fact, our data suggests that perturbation-induced changes in PSI show a strong dependence on the starting PSI value, even when far away from these extreme boundaries (see below). Thus, PSI

values are not appropriate for synergy analysis (and also for linear regression – see our responses to major point 2 of Reviewer #3).

To circumvent this issue, we performed the synergy analysis using our mathematical model which is formulated based on splice isoform ratios. These ratios are not bounded and show the same perturbation-induced fold-change in the splicing outcome irrespective of the starting PSI value (i.e., the nature of other mutations present, see below). Hence, in log-space the perturbations add up, and deviations from additive behaviour between mutations and knockdown can be quantified using a synergy score (**Figure 5b**).

How do we know that isoform ratios show additive behaviour of perturbations in log-space? As detailed in the response to major point 2 of Reviewer #3, we showed this by analysing a system with competing splicing reactions using kinetic modelling (**new Supplementary Figure 7e**; Eqs. 1a and 1b in the Supplementary Material). We validated this prediction by a careful comparison of single vs. double/triple mutation minigenes (**Supplementary Figure 4a**) and by the successful description of the complete mutagenesis dataset using a regression model that is based on splice isoform ratios (see response to major point 2 of Reviewer #3).

In terms of knockdown-mutation synergy, we confirm that – on a global level – *HNRNPH* knockdown induces a very similar fold-change in the AE skipping-to-inclusion ratio across the majority of the minigene library (**Figure 5b**, right). This again corresponds to additivity of mutations and knockdown in log-space. In contrast, when analysed at the level of PSI, the knockdown effect depends in a non-linear manner on the starting PSI values (**Figure 5a**).

We acknowledge that mutations which shift the inclusion frequency close to 0% can be problematic also in our framework. This is due to the fact that the measurements become less accurate at such low levels, i.e., the error in synergy calculations increases. We show this in the **new Supplementary Figure 7g**, in which we plot the uncertainty of the synergy z-score (standard deviation between replicates) as a function of the (inferred) inclusion frequency with a single mutation. In line with instability of results, we find that the z-score standard deviation increases near 0% inclusion. To account for these boundary effects, we now apply an additional filter to exclude mutations from the synergy analysis that on their own completely abolish splicing, and thereby prevent the KD from having additional measurable effects.

Altogether, we conclude that our model that is based on splice isoform ratios provides us with a tool to analyse synergy without strong biases arising from boundary effects.

The fact that 707 mutations (out of 1787) show epistatic interactions with hnRNP H suggest that something is off and that the premise to assume epistasis in all cases may be wrong. Nevertheless, I agree that some known hnRNP H binding sites that have an impact on splicing when mutated in exon 11 seem to display epistatic interactions.

First of all, we have to apologise for accidentally reporting a wrong number of synergistic interactions which led to an overestimation of synergy effects in the original version of the manuscript. With the cutoff at $|z\text{-score}| > 2$ that we initially used, we find a total of 354 mutations in 278 positions that significantly alter the KD response of at least one splicing isoform (instead of 707 as erroneously reported in the original manuscript).

In order to reliably detect synergistic interactions between single mutations and *HNRNPH* KD, we applied stringent thresholds as shown in the original manuscript. Detection required significant *P*-values after multiple testing correction, a mean synergy $|z\text{-score}| > 2$ and a consistent direction of synergy in all replicates. As mentioned above, we obtain 354 mutations in 278 positions with significant synergistic interactions using these original criteria. We believe that it is well possible that a few hundred sites could show synergistic interactions with the *HNRNPH* knockdown, e.g., due to perturbation of higher-order complexes that contain HNRNPH. This is also consistent with the broad HNRNPH binding pattern that we observe in the iCLIP experiments.

In order to highlight the most important synergistic interactions, we now applied additional, more stringent cutoffs. We required the mean synergy score to be $|z\text{-score}| > 3$ or > 5 , identifying 222 and 66 mutations in 184 and 58 positions, respectively. Notably, with increasing thresholds, the significant synergistic interactions strongly focus within the HNRNPH SRBS of cluster 3 in the alternative exon, which capture 42% of the strongest synergistic interactions that affect AE skipping ($|z\text{-score}| > 5$). Thus, by applying stringent cutoffs, our synergy analysis pinpoints the most relevant HNRNPH-sequence interactions.

In the revised version, we now visualise these results by comparing the three applied cutoffs in the revised **Figure 5c** and **Supplementary Figure 12c**. In addition, we corrected and extended **Supplementary Table 4c**, which now provides details on significant interactions (positions and mutations) at the three different cutoffs ($|z\text{-score}| > 2$, > 3 and > 5).

4. Cooperative binding analysis is tantalizing but needs to be better documented by testing other mutations that affect putative H binding sites. As binding of H to G348 does not appear to be affected by G305A, testing G348C should also be tested, as well as other weaker hnRNP H sites and sites that do not respond to KD in exon 11.

We agree with the Reviewer that further mutations should be tested to substantiate this result. We therefore performed the iCLIP experiments including two additional mutations in other SRBS within cluster 3 (G348C and G331C). We also repeated the experiments for the original mutation G305A and the wt RON minigene, performing two replicate experiments for all constructs.

Consistent with our initial observation, each of the mutations triggered a strong reduction of HNRNPH crosslinking not only at the mutated site itself, but also at neighbouring SRBS within cluster 3. This interdependent crosslinking pattern strongly supports our hypothesis that HNRNPH cooperatively binds at multiple SRBS within cluster 3. In the interest of time and resources, we refrained from testing further mutations in other SRBS.

The new iCLIP experiments are shown in **Figure 6a**.

5. I could not find Supplementary Data 2.

We apologise for the inconvenience. An updated version is now provided with the revised version our manuscript.

Reviewer #3 (Remarks to the Author):

Summary:

In this work Braun et al perform extensive mutagenesis screen of RON exon 11 and build a linear model for the effect of the identified mutations on splicing outcome. They measure the accuracy of their model using train/test and comparing it to a strawman which is simply the median effect of a given single point mutation across all mutation combinations in which it was measured. They identify several SF which could be regulating exon 11 using several lines of previous evidence (SF KD, SF CLIP peaks), and correlate exon 11 inclusion to the expression of those in TCGA and GTEX. The most prominent regulator they identify and focus on is hnRNP H, which was previously reported and studies in this context (Ghigna, et al "Cell motility is controlled by SF2/ASF through alternative splicing of the Ron protooncogene". Mol Cell 2005). They KD hnRNP-H and compare the effect of their library of mutations in WT and hnRNP-H KD. They show different hnRNP-H sites have antagonistic and non-linear effects, and its expression levels correlate with exon11 inclusion in TCGA/GTEX. Finally, they show their model predictions for mutation effect on splicing correlate with the changes observed in TCGA cancer patients who harbor these mutations.

The authors should be congratulated for creating such an extensive work that spans many experiments, modeling, and large-scale genomic analysis all tied nicely together. The writing is clear, many of the figures are beautiful and smart. We really liked how the authors went to real data (TCGA/GTEX) to test their findings, compared the effect of the mutations in WT to hnRNP-H KD, and identify complex interactions for hnRNP-H sites. Of note, such analysis was not done in previous highly related work (which is also good) by Julien et al and Rosenberg et al. With that mind, we still found many issues which should be addressed in full.

Major Comments:

We struggled with some of the basic definitions and choices of the predictive model. Some of those were confusingly described, some seem like they might be misguided. Specifically:

1. The authors repeatedly emphasize that they do not model PSI directly/linearly but use ratios instead and that this allows them to avoid issues with an otherwise linear model (paragraph from line 160). Instead, they explain in the supp that they model ratios from a baseline, which yields 5 *separate* regression models. But, as they acknowledge, this gives them a free parameter so they renormalize to get PSI values. So, after all the nice explanations of kinetics (which could serve as a motivation), the math as far as we can tell boils down to be equivalent to what you would get from a softmax regression. (eq. 10,11 in supp). Moreover, they use log PSI, so the natural derivation given what they are trying to model and what they eventually output is a (sparse) logistic regression model using a softmax, and optimizing the conditional log-likelihood, or equivalently the cross entropy. But the authors instead keep those as separate models, optimize each using an L2 distance, and then renormalize. This seems to get them into trouble with what they refer to as nonlinear effects (page 4 bottom in supp) as the info between the separate optimizations is not shared. Running such multiclass (sparse) logistic regression with cross entropy, L1, or L2 should be a straightforward exercise using packages such as scikit-learn. The authors should

definitely try those and if we missed something or these are inferior they should clearly explain/demonstrate this.

As stated by the Reviewers, we perform our regression analysis using isoform ratios (Eq. 7 in the Supplementary Material) and then use our modelling results (i.e., inferred single mutation effects on isoform ratios) to calculate the more intuitive single mutation effects on splice isoform frequencies (related to PSI) by renormalisation (Eqs. 10 and 11). Conceptually, we do not see an advantage of fitting isoform frequencies instead of isoform ratios: as stated in the Supplementary Material, not all six splicing parameters K_i can be determined based on our data, but only five ratios of these parameters (Eq. 4). This non-identifiability of one parameter is not due to our modelling approach, but lies in the nature of our data which consists of six isoform frequencies that sum up to 100%, thus giving only five independent values per minigene. Therefore, a free parameter remains even if the model is fitted directly to the measured isoform frequencies. In return, all six isoform frequencies can be calculated based on the single mutation effects on five isoform ratios with respect to a reference isoform, like we show in Eqs. 10 and 11 in the **Supplementary Material**.

The Reviewers propose to employ a (sparse) logistic regression using a softmax for our analysis. Previously, softmax models were used to infer whether sequence features near alternative exons reduce or enhance splicing in certain tissues relative to the average splicing across all tissues (Barash et al., Nature 2010). In these published models, the response in the training dataset was categorical (enhancement, reduction and no change in PSI), and the model could be used to predict the probabilities of these categorical outputs for new inputs. In our case, the output in the training dataset is continuous and includes six quantified splice isoforms frequencies for each minigene. Nevertheless, a softmax regression can be performed by categorising our data (i.e., using a single splice isoform as output) and using the measured isoform frequencies as sample weights: specifically, six categorical measurements (splice isoform outcomes) were assigned to each minigene and weighted by the measured isoform frequencies. We have implemented such a regression model using cross-entropy loss and L2 regularisation as optimisation functions with the package scikit-learn, as suggested by the Reviewers. We find, however, that the results are slightly inferior compared to our original method: While the softmax shows comparable efficiency in fitting the data, it performs less well than our ratio-based approach when predicting minigenes that had not been part of the training dataset in a 10-fold cross-validation (**new Supplementary Figure 7a**).

From the Reviewers' comment (and also point 3 of Reviewer #2), it became clear that the usefulness of regression based on splice isoform ratios as opposed to PSI (or related metrics) did not become entirely clear. To clarify this issue, we also performed an alternative regression in which we assume additivity of mutation effects at the level of isoform frequencies, and then use these isoform frequency equations directly for regression (while constraining the sum of frequencies for each single mutation to one). Using this approach, we find that the fit quality and especially the predictions in the 10-fold cross-validation were also worse compared to our ratio-based approach, as mentioned in the revised manuscript and shown in the **new Supplemental Figure 7b**.

The Reviewers comment that the use of isoform ratios gets us into trouble with nonlinear effects. In fact, we think that the use of isoform ratios is essential to avoid nonlinearity issues

(see also a detailed description of our definition of linearity in our response to major point 2): In short, a mutation only elicits the same fold-change, irrespective of the mutational background, at the level of isoform ratios, whereas this is not true for the bounded PSI metric for which the same mutation induces different fold-changes near 0% or 100% (Eq. 4 in **Supplementary Material** and **Supplementary Figures 3a and 7e**). Hence, in log-space, mutation effects add up in terms of isoform ratios, but not in terms of the bounded PSI metric, implying that only the former are suitable for linear regression.

In the Supplementary Material, we stated that minigenes which simultaneously contain mutations at two splice sites had to be left out due to nonlinearity. We would like to point out that this is not a problem specific to our model: The maximally possible reduction of inclusion is already reached by a single splice site mutation, since minigenes which harbour an additional mutation in a second splice site, show a very similar distribution of inclusion frequencies (**new Supplementary Figure 7f**). Alternative regression models will also not capture that splice site mutation effects cancel out, implying that this problem is not specific to our approach. We clarify this issue in the revised version of the **Supplementary Material**.

Taken together, we continue to believe that our isoform-ratio modelling approach is superior compared to alternatives based on PSI or similar metrics, as we explicitly take into account competition effects in splicing and boundary effects due to normalisation.

2. The issue about what is linear or not and what they compare to affects their claims/statements in the discussion as well regarding what the model is doing (lines 428-433). It will also affect their discussion of linearity and related work in lines 434-445. Specifically when they state the “majority” is not linear (line 436) it’s not clear what is that “majority” and again, what is the definition of this linearity. If they took out the fraction of close proximity mutations done in Julien et al 2016 would their results agree, or is it also a result of how they defined “linearity”? Also note that Rosenberg et al used a quite different model over k-mers, something not mentioned.

We define linearity based on the way the mutational effects cumulate in minigenes exhibiting several mutations. Specifically, we termed a mutation effect linear if a mutation induces the same fold-change in a splicing outcome irrespective of the mutational background (i.e., the nature of other mutations present). If such linearity is fulfilled the mutation effects add up in log-space and a linear regression can be performed to infer single mutation effects from the measured combined mutations. We added a more precise definition of linearity to the revised manuscript and explain under which circumstances such linearity can be assumed in the following paragraph.

Using kinetic modelling, we had analysed a system with competing splicing reactions *in silico* in the original manuscript (Eqs. 1a and 1b in the **Supplementary Material**) and found that splice isoform ratios show proportional (in our sense linear) changes to mutations affecting splicing kinetics, irrespective of the presence of other mutations (**new Supplementary Figure 7e**). In contrast, mutation-induced fold-changes depend on the mutational background at level of splice isoform frequencies (and are thus nonlinear in our notation) (**new Supplementary Figure 7e**). To test for such linearity in our data, we had analysed mutations that are present as single mutation minigenes in our library and simultaneously occur in combinations as double/triple mutation minigenes (**Supplementary Figure 4a**). At

the level of splice isoform ratios, we found that the log fold-change in double/triple-mutation minigenes corresponds to the sum of the corresponding single mutation log-fold changes, as predicted by the kinetic model. To more generally confirm the linearity of mutations in log-space, we fitted a linear regression model that is based in splice isoform ratios to the complete dataset in the original manuscript (**Figure 2b and Supplementary Figure 5b**). We find that 94% of the splice isoform frequencies are fitted within 5% deviation from the measured value. Thus, the vast majority of combined mutation effects in the dataset can be described based on the assumption of additive (linear) single mutation effects.

To support our initial statement of predominantly linear interactions between *cis*-regulatory elements (which was questioned by the Reviewers), we turned to the subset of 142 minigenes that contain two or more simultaneous mutations in splice-regulatory binding sites (SRBS) of HNRNPH. As a measure of linear mutation interactions, we analysed the goodness-of-fit of our regression model and found that the minigenes containing two simultaneous HNRNPH SRBS mutations had a goodness-of-fit comparable to the complete minigene population. Specifically, 93% of the splice isoform frequencies in this set can be explained within 5% deviation from the measured value. Thus, cooperative (nonlinear) interactions rarely occur if a minigene harbours effective mutations within two *cis*-regulatory elements (**new Supplementary Figure 15e**). However, among the 18 minigenes containing simultaneous mutations in two HNRNPH sites *within the alternative exon* (cluster 3), several were fitted worse than the remainder of the minigene population (see comment 5 below for more details), suggesting that a deviation of our model from the data can be indicative for exceptional cases of cooperative mutation interactions (**new Supplementary Figure 15e**). We mention these findings in the revised Discussion.

To address the Reviewers' comment that nonlinearity (cooperativity) between mutations may specifically arise for close-proximity mutations, we related the goodness-of-fit of our linear regression model to the nearest distance between two effective mutations in each minigene. To this end, we calculated the fitting error by summing up the residuals over all splice isoforms, and defined significant mutation effects as described in Methods (in total 778/1022 mutations with significant effects in at least one isoform for HEK293T/MCF7 cells, respectively). We found no clear effect of the mutation proximity on the fitting error, as judged by the median fitting error over all minigenes harbouring the same mutation distance (**new Supplementary Figure 7c**). Notably, 1682 (HEK293T)/2344 (MCF7) minigenes in our screen contained at least two splicing-effective mutations but only 84 (HEK293T)/139 (MCF7) of them had a nearest effective mutation distance of less than seven nucleotides (used here as an arbitrary cutoff for close proximity; **new Supplementary Figure 7d**). For both cell lines, we herewith cover only 1.3% of all theoretically close-proximity double mutations possible in our minigene (each single mutation being effective on its own). This indicates that our dataset does not exhibit enough coverage to detect cooperative interactions of nearby mutations.

We added the information contained in the last two paragraphs to the revised manuscript.

*

3. The authors exclude ~6% of the variants as they can not model those, even though they acknowledge these can have a significant effect and create many variants (line 141, page 5

top in supp). Since they are interested in modeling cancer-specific variations it might very well be that those 6% are particularly important/deleterious. What are the fractions of those unmodeled in the cancer data? Do they have a significant effect based on their metric even without a model and correlation to what happens in cancer (e.g. median over all variants including those?).

We agree with the Reviewers that mutations excluded from the model might include particularly deleterious instances. However, comparison to tumour data (COSMIC and TCGA) showed that none of these mutations has been reported in cancer. Nevertheless, they might have pronounced effects on splicing in our screen. We therefore analysed them in more detail, and provide their effects (as judged by the median-based metric) in **Supplemental Table 3**.

As detailed in the Supplementary Material (page 5f), we excluded certain minigenes from the linear regression model. These harboured either (i) combinations of two splice site mutations or (ii) showed a strong activation of cryptic splice sites (new splicing products ('other') behave very heterogeneously). In MCF7 cells, we exclude on average of 5% of the minigene variants per replicate (**Supplementary Table 2**). In total, this results in 97 mutations that are removed in all three replicates, including 11 splice site mutations and 86 mutations in other 78 positions in the minigene.

Moreover, while the information about these mutations is lost using our model, we could infer the effect of other types of mutations at the same positions. This means that the corresponding positions are not lost entirely, and still considered for the identification of splicing-effective *cis*-regulatory elements (SRBS). Specifically, in MCF7 cells, 63 of the 78 positions with excluded mutations are backed by other mutations with a significant splicing change (>5% change in any isoform). Importantly, these include all positions in HNRNPH SRBS for which we had to exclude certain mutations.

Taken together, we conclude that the loss of information due to exclusion of certain mutations is moderate for the *RON* minigene. This may, however, not be true in future random mutagenesis screens of other minigenes. For such cases, mutations giving rise to non-canonical splice isoforms need to be described in more detail in the linear regression analysis, e.g., by treating each 'other' variant as a separate variable in the model.

Related to that, the model can not handle the 608 short insertions and deletions (line 684) which are ~3% of the variants they created, but this model limitation is not discussed in the main text.

We are sorry for this misunderstanding. As stated in the Methods section, the 608 insertions and deletions in our mutated minigene library were taken into account during the modelling step. They were treated as independent mutations in addition to the canonical single nucleotide variants, and hence their information was exploited for inferring single mutation effects from the library.

Insertions and deletions are conceptually different from single nucleotide variants in terms of their functional analysis and interpretation. Moreover, the error-prone PCR used for the random mutagenesis is optimised towards single nucleotide variants. As a consequence, the few present insertions/deletions are repeated in only very few plasmids, making conclusions

on their potential splicing effects less accurate and reliable. In the present manuscript, we therefore decided to focus on the single nucleotide variants for in-depth analysis.

In order to make the information on insertions/deletions available for further analyses, we now include the model-inferred splicing effects of all measured insertions and deletions in the revised **Supplementary Table 3**. An additional column indicates the type of mutation to facilitate the extraction of insertions/deletions. In addition, we rephrased the misleading sentence in the Methods section.

*

4. The fact they don't find increased conservation of regulatory elements in the introns (line 235) is not well explained/discussed. This is also in clear contrast to many other works that show highly conserved intronic regulatory elements for splicing. One possible explanation would be that the intronic positions they identified exhibit a smaller effect (do they? You could check!) and/or that the previous works focused more on highly conserved tissue-specific regulation while in this case there are just general (subtle?) effects on the base level of inclusion. Either way, authors should extend the discussion here.

Following the Reviewers' comment, we re-evaluated the phylogenetic conservation and realised that a mix-up had occurred in the initial analysis. We apologise for this mistake. The corrected plot is shown in the revised **Supplementary Figure 9c**.

In the corrected analysis, we find that splicing-regulatory positions within introns show significantly higher conservation than splicing-neutral positions in the same regions. As the Reviewers mentioned, this is in line with several previous works, and we now discuss it accordingly. In the exons, the situation is different, most likely since phylogenetic conservation is generally much higher in these regions.

As suggested by the Reviewers, we compared effect sizes across the different regions in our minigene. Consistent with previous work, the strongest effect sizes are clearly found in the alternative exon and the flanking introns, and decrease towards the constitutive exons. This can also be nicely seen in the full profiles in Supplementary Data 2 & 3. Nevertheless, as detailed in the response to comment 2 of Reviewer #2, we detect a substantial number of very strong mutations in all regions of the minigene.

In the revised version of our manuscript, the effect sizes per transcript region are shown in the **new Supplementary Figure 9d**. We additionally stratified mutations by their effect size in the comparison of phylogenetic conservation scores in **Supplementary Figure 9c**. Moreover, as detailed above, we now report and visualise the splicing-effective mutations at different cutoffs (**Supplementary Figure 8** and **Supplementary Table 4**).

*

5. The results on cooperative hnRNP H sites leading to a splicing switch are very nice. The authors gloss over whether their mutation model is able to capture that or not, which should be noted. Also, this result should be noted as in line with previous experimental results and splicing code predictions made for CU/CUG elements around daam1 exon 16 (Barash et al Nat 2010).

Our linear regression model does not capture cooperativity between mutations or binding sites. As stated in our response to major point 2 above, we show that – on a global scale – such cooperativity can be neglected as judged by the good model fit to the data. To more specifically address the Reviewers' comment, we checked the set of co-occurring mutations in our screen and found that we have 18 minigenes exhibiting two simultaneous mutations within cooperative HNRNPH binding sites (303-312, 327-333 and 347-351) in the alternative exon. On average, these minigenes are fitted worse than expected: 7 (39%) have a fitting error > 0.2 , compared to 9% in the set of all fitted minigenes with at least two effective mutations (**new Supplementary Figure 15e**). This supports the existence of cooperative behaviour between HNRNPH sites in the alternative exon and suggests that a deviation of our non-cooperative model from the data can be indicative for cooperative interactions between mutations. We mention this in the revised Discussion. Furthermore, we cite Barash et al. in the context of cooperative interactions between *cis*-regulatory elements controlling splicing (see revised Discussion).

*

6. We found the use of the term “epistatic” interactions between hnRNP H KD and mutations to be extremely confusing. In general, the term is commonly used for the non-additive effect of genetic variations, but not here. Furthermore, the epistatic interaction here happens when a mutation in WT behaves differently than under hnRNP H KD. So “normal” interaction i.e. when it's a BS for hnRNP H, are now denoted “epistatic”! Worst, the term is used differently in Julien et al for variant combinations, which makes it all the more confusing. Why??

As stated by the Reviewers, we use the term epistatic “when a mutation in WT behaves differently than under hnRNP H KD”. Thus, we refer to a non-additive effect of two perturbations, KD and mutation, in similarity to the use in the literature. However, we agree that the term may be confusing given that it was used differently in Julien et al., and mostly refers to combined mutations in other studies.

In response to the Reviewer's concern, we therefore now refer to the epistasis effects as ‘synergistic interactions’. Synergy can be defined as a nonlinear relationship between two or more elements whereby they generate a combined outcome that is more or less than the sum of their parts taken separately, due to their capacity to work together or against each other. (“Synergy Types,” *Complexity Labs*, <http://complexitylabs.io/positive-negative-synergies/>).

In the context of our analysis, a negative synergistic interaction (z -score < 0) of a given mutation would be a reduced response to *HNRNPH* knockdown compared to ctrl, i.e. a mutation that weakens an HNRNPH binding site. In contrast, a positive synergistic interaction (z -score > 0) points to a mutation that strengthens or generates a new HNRNPH binding site.

*

7. The authors gloss over the fact that their results show “epistatic” effect for hnRNP-H for 468/550 positions (!!) in MCF7 cells. This is worth explaining/discussing.

Please note that the following explanation already uses the term ‘synergistic interactions’ (see above).

As mentioned in response to comment 3 of Reviewer #2 above, the high number of synergistic interactions was a mistake in the original manuscript. We apologise for this and corrected the manuscript accordingly.

For the original cutoff at an absolute synergy score of $|z\text{-score}| > 2$, the correct number are 354 mutations in 278 positions which display significant synergistic interactions. In addition, we applied more stringent thresholds to support that our synergy analysis reliably detects HNRNPH-RNA sequence interactions and to detect the most relevant HNRNPH SRBS. In brief, using a cutoff of $|z\text{-score}| > 5$, we find that 42% of the synergistic interactions for AE skipping (22 out of 52 mutations) fall into the SRBS cluster 3 in the alternative exon.

Figure specific comments:

=====

1. Fig2B:

This figure you can't see much. Could plot it separately for each variant at least in the supp. They could also plot for each variant the max deviation in PSI so we see how good/bad are the variants prediction e.g. X% of them vary more than Y% PSI than the observed.

In addition to the plot in Fig. 2b, we now provide separate plots in the **new Supplementary Figure 5a**. We also state in the revised Discussion that 94% of the fitted isoform frequencies are within 5% deviation from the measured value. We use this quantitative information as an argument for our conclusion that most mutations interact linearly (non-cooperatively) in the *RON* minigene (see response to major comment 2 above). Following the Reviewers' suggestion, we also added the **new Supplementary Fig. 5b**, in which we show the fraction of minigenes fitted within a certain deviation between model and data as a cumulative histogram, separately for each splice isoform.

2. Fig2C:

This is a nice result and analysis. But it would be good to know (a) what is the distribution of #occurrences i.e. the PDF/CDF of mutation occurrences in the data (the x-axis here) (b) combining the above and their current graph, compare to a median (strawman) how many (CDF) the improve by more than X% in error compared to the strawman. This would give a sense of what is the overall gain in accuracy from the model compare to only using the experimental data.

Following the Reviewers' comment, we now provide three additional plots to support this analysis: (i) **Supplementary Fig. 2e** shows the distribution of mutation occurrences in the data, showing that more than 50% of the mutations in the library occur in at least four different minigene variants. (ii) **Supplementary Fig. 5c** shows the number of mutation occurrences in the leave-one-'single-mutation-minigene'-out cross-validation procedure as a histogram. This analysis demonstrates that a high number of tests (>100) were available for mutation occurrences <20, thus supporting that our cross-validation results are robust and reliable. (iii) As suggested by the Reviewers, we visualised the overall gain in accuracy from

the model (**Supplementary Fig. 5d**). To this end, we generated a cumulative histogram, which shows how AE inclusion absolute errors (%) differ between the model and median-based estimation (x-axis) for the leave-one-“single-mutation-minigene”-out cross-validation tests (y-axis). In 65% of the tests, the model captures the measured single-mutation effects better than the median-based prediction. The model is also more accurate in absolute terms: When the median outperforms the model (35%), it only leads to a minor improvement in accuracy. In contrast, the gain-of-accuracy by the model is more pronounced, and can improve the AE inclusion inference by as much as 50%, especially when a mutation is rare in the dataset (low mutation occurrences).

3. Fig3C:

This result is somewhat disappointing, and its presentation both here and in the main text is misleading. In practice, the r correlation value hinges on two clear outliers which have a strong effect, G370T and G297A. They should report the correlation without these as well and discuss this. Also, G297A is a splice site one so not very interesting. The G370T is more interesting, but is it in an hnRNP H site? something else? it is not even shown in Fig B for some reason.

We agree with the Reviewers that the correlation in Figure 3c is driven by the two strongest mutations, i.e. G297A and G370T. As suggested, we now additionally report the correlation without these two data points which is reduced and no longer significant (Pearson correlation coefficient $r = 0.27$, P-value = 0.12). One explanation for this loss in correlation is that the TCGA data display a high level of inherent variability. We find that control patient samples without a given mutation display an average standard deviation in *RON* exon 11 skipping of 14%. This is also apparent in Figure 3d in which the samples without mutation cover a wide range of splicing levels. Taken together, most mutations in Figure 3c display small effects in the range of one control standard deviation, which makes quantitative comparisons with our screen difficult. Notably, however, the two mutations with strong effects (G297A and G370T) show an excellent quantitative agreement with our screen. This observation supports the notion that our high-throughput screen recapitulates strong *in vivo* splicing changes in human cancers.

In order to visualise the inherent variability in the TCGA splicing measurements, we included the standard deviation of unmutated samples in the revised version of **Figure 3c**.

We agree with the Reviewers that mutation G370T is an interesting example. The mutation does not fall into an HNRNPH binding site. It overlaps with predicted splice-regulatory binding sites of QKI and HNRNPL (**Supplementary Figure 10**), but expression of neither of the two proteins correlates with *RON* exon 11 splicing in TCGA or GTEx (**Supplementary Table 6**). Given that the same mutation also introduces a premature stop codon at the protein level, the newly gained information about its splicing-regulatory role offers a completely new interpretation of its molecular consequences. This observation nicely illustrates the power of our screen to detect splicing changes of potential relevance in cancer. Our screen thereby adds to the increasing evidence that synonymous and non-synonymous mutations can significantly alter alternative splicing and thereby have deleterious impact in cancer (Gartner et al., 2013; Supek et al., 2014; Gotea et al., 2015; Jung et al., 2015).

Please note that Figure 3b shows mutational information that was retrieved from a different database (COSMIC). Mutations that are identified in TCGA tumour cohorts should eventually be submitted to COSMIC, but may appear with a certain delay as data need to pass curation before being added. For the revised version, we confirmed that mutation G370T is currently not yet present in the COSMIC database.

4. To show the utility of their model they should compute and report the same correlation if they used just the median value for each of those mutations. This will show the relative gain of their model compared to using the experimental results directly.

As suggested by the Reviewers, we utilised the TCGA analysis (Figure 3c) to compare the performance of our model-inferred mutation effects to the simpler estimation based on median effects of all minigenes with a given mutation. As shown below (Figure III a,b), the correlation against TCGA data is virtually identical for the model-inferred and the median-based single mutation effects. Consistently, the values inferred for these mutations by both methods are highly correlated (Figure III c). One reason why there is no visible improvement with the model-inferred values might be that most of these mutations have very good plasmid support in our minigene library, thereby strengthening the median-based estimation. Moreover, most mutations in this analysis show only minor splicing changes and are hence more susceptible to the inherent noise in the TCGA data and our screen (see also comment figure-specific comment 3 above).

Figure III: Analysis as in Fig. 3c, assessing the performance of model-inferred mutation effects (a) versus median-based estimates (b).

Given that we show earlier in the manuscript that the model outperforms the median (Figure 2c and also in the new Supplementary Figure 5d), especially at low mutation occurrences, we decided to not return to a comparison of both methods in the context of the TCGA patient data in the revised manuscript.

5. Fig3E:

This is a nice connection but it is not fully explored. The pvalues on the right are somewhat misleading as they relate to the effect measured of KD in a completely different work. What would be much more informative would be to see the correspondence between their model prediction for taking away the sites of a SF and the KD experiment effect. Of course, a site

could be shared, nonetheless, it would indicate how well they can perform such an analysis with their model.

We agree with the Reviewers that it would be informative to directly connect the mutation effects in our screen with the RBP KD effects in Papasaikas et al, 2015. However, this is not trivial, not only because a binding site could be shared, but also because many RBPs are predicted to bind at multiple binding sites. As these binding sites often lie in different transcript regions and are hence not unlikely to differently impact on splicing, it is unclear how their individual effects will integrate into a unified splicing response.

The complex regulatory integration of multiple binding sites is exemplified by our prime regulator HNRNPH which shows a total of 22 SRBS across the *RON* minigene region. The majority of these sites are indeed bound by HNRNPH (iCLIP data, Figure 4a) and show opposing effects on *RON* exon 11 inclusion (mutation analysis, Figure 4c). For instance, binding at intronic SRBS cluster 2 promotes AE inclusion, whereas cooperative binding a multiple SRBS in the alternative exon leads to strong AE skipping. Although certain positional rules have been previously established for splicing regulation, we believe that a detailed analysis of this is beyond the scope of this manuscript.

In order to address the Reviewers' concern, we revised **Figure 3e**: To visualise which mutations are in line with the published KD effect, we colour-coded the predicted binding sites according to whether the majority of mutation effects within a binding site agree with the direction of the published RBP KD effect. This highlights the most relevant SRBS for each RBP, and therefore provides a resource for follow-up studies investigating RBP effects on *RON* exon 11 splicing.

Minor Comments:

=====

In various places, results/conclusions are overstated. For example:

The title is somewhat misleading as it implies decoding of the entire *RON* gene, in practice only for exon 11.

In line with the Reviewers' comment, we changed the title to "Decoding a cancer-relevant splicing decision in the *RON* proto-oncogene using high-throughput mutagenesis".

The abstract overstates the results, should be toned down. We don't see the results, as presented, offering "insights on the functional impact in human disease".

We agree with the Reviewers and changed the sentence to: "Our results thereby offer new insights into splicing regulation and the impact of mutations on alternative splicing in cancer."

Similarly, at the end of the intro, the authors state: "Our approach promises new insights into the molecular function of cancer-associated mutations and the mechanisms of alternative splicing regulation in general." Again, this seems like an overstatement given

what is actually delivered, which only applies (with limited capacity, as discussed above) to exon 11 in RON.

We refer here to the mutagenesis screening approach in general and not to our *RON* exon 11-specific data. We changed the sentence to: "Our mutagenesis screening approach promises new insights into the splicing effects of mutations in humans and the mechanisms of alternative splicing regulation in general."

In line 421 in the discussion, authors summarize that their model "can reliably predict the 421 individual effects of almost 1,800 mutations". This statement is misleading: The model is not able to predict the effect of *any* mutation it has never seen. It can infer the effect of a single mutation given several combinations in a manner which is more accurate than trying to simply guess by the mean/median.

We agree that the term 'predict' was misleading and revised the complete main text, figures and supplementary material accordingly. As suggested by the Reviewer we now use 'infer' instead of predict.

In lines 475-478 the authors claim their method can be extended further enable to reconstruct complete splicing code. Given the usage of tailored mini-gene per exon/gene and KD of each SF for each of those, it seems this claim for large-scale analysis should be toned down or removed.

We toned down the statement which now reads:

"In the future, this approach can be used to incorporate further RBPs to dissect the regulatory network of selected splicing decisions."

Line 480 states the study shows hnRNP-H is a "master" regulator. Why? "Key" or "important" seems more appropriate here.

We rephrased the sentence accordingly, now using the term key.

*

References choices in the intro seem to reflect some knowledge of recent literature but are not aligned well with the statements and background. For example, Yang et al 2016 is relevant for showing the effect on protein-protein interaction, but this was shown well before that on a large scale in Ellis et al Mol Cell 2012. The observation that over 90% of human genes undergo AS should be attributed to the back to back papers Pan et al Nat Gen 2008 and Wang et al Nat 2008. Similarly, the term 'splicing code' is usually attributed to the Wang and Burge RNA 2008 review that defined what is such a code and specified it as a long-term goal, or to Barash et. al Nat 2010 which offered the first such code derivation.

We included the suggested references.

*

Just as the references seem lacking, the authors do not draw connections to numerous previous works. For example:

In line 214 they report that they were surprised to find the splice site of downstream exon12 affects the skipping of exon 11. The strength of the up/down sites were already included in the original splicing code as an informative feature (Barash et al, Nat 2010) and numerous other works identified binding SF around those as affecting ES (e.g. Llorian et al NSMB 2011) so the result should not be such a surprise.

We agree that exon definition had been described before and therefore changed the sentence to: "Consistent with an exon definition model of splicing we find that..". We also refer to the suggested publications.

Line 486-489 again this conclusion is in line with numerous works, from classical SrcN1 works by Doug Black mapping it's regulatory elements in a series of papers, to more computational works as in Barash et. al Nat 2010.

We agree with the Reviewers that several of our observations regarding *RON* exon 11 splicing have been made in the context of other splicing events. However, due to the limit in the number of references in *Nature Communications* we decided not to include additional citations to this conclusion.

By far the most relevant work is Julien et al. The authors do mention it several times, but given the strong connection it would help readers to more directly compare/contrast the various conclusions/results for each section. This should not be seen as taking away from the impact/relevant of this paper as there is a lot of unique results in this paper already (see opening statement).

We fully agree with the Reviewers that the very nice work of Julien et al. is of particular interest to our study. As pointed out by the Reviewers, we already cite Julien et al. several times. In particular, we already discuss the most relevant findings of this work (in particular the dense mutational landscape and extensive epistasis between mutations) in the context of our data. Therefore, due to the limit in space we decided not to extend these aspects.

REVIEWERS' COMMENTS:

Reviewer #1 (Remarks to the Author):

In their revised manuscript, the authors have adequately addressed all reviewer comments with new experimental data

Therefore, I have no additional reservation.

Reviewer #2 (Remarks to the Author):

The authors have addressed to my satisfaction most of the criticisms and comments. I am enthusiastically recommend it to be accepted for publication

Reviewer #3 (Remarks to the Author):

The authors did a comprehensive job addressing many of the reviewers' comments. I enjoyed reading the other reviewers thoughtful comments and suggestions as well as the responses to those. Overall, the authors added experiments (overexpression, iCLIP with mutations); additional analysis such as the softmax regression and relation to tumor stage; adequately clarified aspects of previous work (linear vs. non-linear effects, differences from Julian et al results, epistasis etc.). Authors also noted a few mistakes that were fixed.

Overall, the revised manuscript is clearer, more precise, and easy to follow. I find this to be excellent work and look forward to seeing it in print.

The only completely minor comment I have is that the term epistasis is still used in supp Table S3.